# TRUSTJUDGE: INCONSISTENCIES OF LLM-AS-A-JUDGE AND HOW TO ALLEVIATE THEM

**Yidong Wang**[1*] **Yunze Song**[2*] **Tingyuan Zhu**[3] **Xuanwang Zhang**[4]
**Zhuohao Yu**[1] **Hao Chen**[5] **Chiyu Song**[6] **Qiufeng Wang**[7]
**Zhen Wu**[4] **Xinyu Dai**[4] **Yue Zhang**[6] **Cunxiang Wang**[8†] **Wei Ye**[1†] **Shikun Zhang**[1†]

[1]Peking University    [2]National University of Singapore    [3]Institute of Science Tokyo
[4]Nanjing University    [5]Carnegie Mellon University    [6]Westlake University
[7]Southeast University    [8]Tsinghua University

## ABSTRACT

The adoption of Large Language Models (LLMs) as automated evaluators (LLM-as-a-judge) has revealed critical inconsistencies in current evaluation frameworks. We identify two fundamental types of inconsistencies: (1) *Score-Comparison Inconsistency*, where lower-rated responses outperform higher-scored ones in pairwise comparisons, and (2) *Pairwise Transitivity Inconsistency*, manifested through circular preference chains ($A > B > C > A$) and equivalence contradictions ($A = B = C \neq A$). We argue that these issues come from information loss in discrete rating systems and ambiguous tie judgments during pairwise evaluation. We propose **TrustJudge**, a probabilistic framework that addresses these limitations through two key innovations: 1) *distribution-sensitive scoring* that computes continuous expectations from discrete rating probabilities, preserving information entropy for more precise scoring, and 2) *likelihood-aware aggregation* that resolves transitivity violations using bidirectional preference probabilities or perplexity. We also formalize the theoretical limitations of current LLM-as-a-judge frameworks and demonstrate how TrustJudge's components overcome them. When evaluated with Llama-3.1-70B-Instruct as judge using our dataset, TrustJudge reduces Score-Comparison inconsistency by 8.43% (from 23.32% to 14.89%) and Pairwise Transitivity inconsistency by 10.82% (from 15.22% to 4.40%), while maintaining higher evaluation accuracy. Our work provides the first systematic analysis of evaluation framework inconsistencies in LLM-as-a-judge paradigms, offering both theoretical insights and practical solutions for reliable automated assessment. The framework demonstrates consistent improvements across various model architectures and scales, enabling more trustworthy LLM evaluation without requiring additional training or human annotations. The codes can be found at https://github.com/TrustJudge/TrustJudge.

## 1 INTRODUCTION

The widespread adoption of LLM-as-a-judge approaches has offered a scalable and effective alternative to costly human assessments Chang et al. (2024); Fu et al. (2023); Lin & Chen (2023); Sottana et al. (2023); Huang et al. (2024); Koutcheme et al. (2024); Song et al. (2024); Zhu et al. (2023). Beyond evaluation, this paradigm also actively contributes to model improvement, enabling self-refinement through iterative feedback Yuan et al.; Wu et al. (2024); Wang et al. (2025c) and collaborative progress via mutual assessment Wang et al.; Li et al.. These LLM-as-a-judge frameworks typically implement two evaluation protocols Li et al. (2023); Chen et al. (2024a); Li et al. (2025); Chen et al. (2024b); Tan et al. (2024); Thakur et al. (2024); Szymanski et al. (2025); Raju et al. (2024)n: *single-score assessment*, where a judge LLM (either general-purpose or specifically fine-tuned for evaluation) assigns integer numerical ratings to model outputs Zheng et al. (2023); Wang et al. (2024), and *pairwise comparison*, where the judge evaluates competing responses in

---

*Equal contribution.
†Correspondence: wangcunxiang303@gmail.com, wye@pku.edu.cn, zhangsk@pku.edu.cn.

direct comparison (with the order of responses swapped in two separate evaluations to eliminate position bias) to produce preference judgments Dubois et al. (2024); Wang et al.; Li et al. (2024).

However, our research identifies two critical inconsistencies in these LLM-as-a-judge evaluation frameworks for LLMs: (1) **Score-Comparison Inconsistency** between single-score and pairwise comparison assessment, where LLMs with lower absolute scores may outperform higher-scored counterparts in pairwise comparisons ($R_x \succ R_y$ despite $score(R_x) < score(R_y)$)[1]; and (2) **Pairwise Transitivity Inconsistency** in pairwise comparison evaluation, where judgments show non-transitive cycles ($R_x \succ R_y \succ R_z \succ R_x$) and equivalence contradictions ($R_x \equiv R_y \equiv R_z \neq R_x$), violating rational preference principles. While prior work addresses pairwise inconsistencies through complex mathematical modeling Xu et al.; Zhang et al., such continual training risks compromising model generalizability Luo et al. (2023); Lin et al. (2024) without resolving score-comparison conflicts. To the best of our knowledge, this is the first work to systematically expose both inconsistencies as foundational weaknesses in LLM-as-a-judge frameworks and to provide a unified resolution.

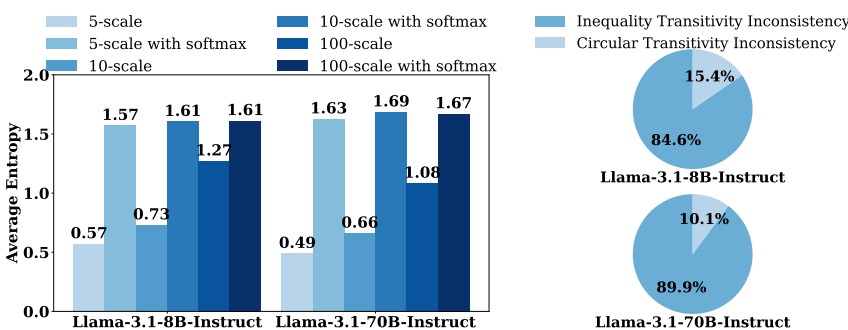

Figure 1: Left: Average entropy of Llama-3 Grattafiori et al. (2024) models' single-score outputs over six rounds on 1,200 instructions, by scoring strategy. Right: Breakdown of circular- vs. inequality-transitivity errors in pairwise-comparison tests.

To address these inconsistencies, we introduce **TrustJudge**, a probabilistic evaluation framework that preserves judgment entropy while resolving both (a) score-comparison conflicts via distribution-sensitive scoring and (b) transitivity violations through likelihood-aware aggregation. As shown in Figure 1, we argue that the score-comparison inconsistency primarily stems from information loss in the integer scoring system—the coarse-grained 5-point scale compresses nuanced quality differences into identical scores (e.g., different quality responses both receiving 4 points), resulting in low entropy judgments that fail to discriminate actual performance gaps. We propose two effective solutions: (1) increasing scoring granularity (5→10→100 points) to preserve more discriminative information, and (2) probabilistic scoring that maintains the full entropy of model judgments. For pairwise transitivity inconsistency, we find most of inconsistencies originate from tie judgments (equivalence contradictions). We propose breaking ambiguous ties by either (1) preferring responses whose entire sentence exhibits lower perplexity, or (2) deciding preference based on the judge's token-level confidence for win, tie and lose. These approaches significantly reduce inconsistency rates while maintaining the scalability and effectiveness of LLM-as-a-judge frameworks. While recent works like Liu et al. (2023); Wang et al. (2025b) adopt probabilistic scoring to enhance human alignment, our motivation differs fundamentally in addressing fundamental inconsistencies of evaluation frameworks rather than improving human-judge agreement. Our probabilistic scoring serves as an entropy-preserving mechanism with granularity enhancement to resolve score-comparison conflicts. Our theoretical analysis shows that discrete scoring systems suffer from information loss by showing that distinct response distributions can yield identical scores despite different entropies, whereas TrustJudge's distribution-sensitive scoring preserves and distinguishes these differences, and further demonstrates that its PPL-based confidence distribution reduces uncertainty in ambiguous cases by leveraging perplexity to create a lower-entropy signal for decision-making.

Extensive experimental results across multiple model families (Llama-3, GPT, Qwen, Gemma) and scales (3B to 70B parameters) demonstrate TrustJudge's effectiveness. Our framework reduces

---

[1]We use $R_x, R_y, R_z$ to represent distinct LLM responses, $\succ$: strictly preferred; $\prec$: strictly worse; $\succeq$: preferred or equivalent; $\preceq$: worse or equivalent; $\equiv$: equivalent; $\neq$: not equivalent

Score-Comparison inconsistency by 8.43% (from 23.32% to 14.89%) and Pairwise Transitivity inconsistency by 10.82% (from 15.22% to 4.40%) when using Llama-3.1-70B-Instruct as judge. These improvements are achieved while maintaining or improving evaluation accuracy, with exact match rates increasing by 1.19-6.85% across different model sizes. Our ablation studies confirm that both distribution-sensitive scoring and likelihood-aware aggregation contribute significantly to these improvements, with the full framework achieving the best performance across all tested scenarios.

In conclusion, we present the first systematic analysis of fundamental inconsistencies in LLM-as-a-judge evaluation frameworks, identifying two critical limitations: (1) information loss in discrete scoring systems causing Score-Comparison Inconsistency, and (2) ambiguous tie judgments leading to Pairwise Transitivity Inconsistency. TrustJudge addresses these through distribution-sensitive scoring that preserves judgment entropy and likelihood-aware aggregation to break ambiguous ties. Our experiments demonstrate significant inconsistency reductions while maintaining evaluation accuracy across diverse models and tasks. This work provides both insights into LLM evaluation limitations and practical solutions for more reliable automated assessment, enabling more trustworthy deployment of LLM-as-a-judge paradigms in research and applications. In addition, we provide a more detailed review of related work in Appendix A.

## 2 METHODOLOGY

Our framework addresses two fundamental inconsistencies in LLM-as-a-judge systems through formal definitions and quantitative metrics. We first establish mathematical characterizations of these inconsistencies, then present our TrustJudge algorithm.

### 2.1 INCONSISTENCY DEFINITIONS AND METRICS

**Definition 2.1** (Score-Comparison Inconsistency). For responses $R_x, R_y$ with scores $S_x, S_y \in \mathbb{Z}$ (e.g., 1-5 scale) and pairwise comparison $C = C(R_x, R_y) \in \{-1, 0, 1\}$ (1: $R_x$ preferred, -1: $R_y$ preferred, 0: tie), inconsistency occurs when:

$$(S_x > S_y \wedge C \leq 0) \vee (S_x < S_y \wedge C \geq 0) \vee (S_x = S_y \wedge C \neq 0). \tag{1}$$

The **Conflict Ratio** $CR = \frac{1}{n} \sum_{i=1}^{n} \mathbb{I}[\text{inconsistent pair } i]$ measures prevalence, where $n$ is total pair numbers in the test set and $\mathbb{I}[\cdot]$ is the indicator function.

**Definition 2.2** (Pairwise Transitivity Inconsistency). For a set of $n$ responses $\mathbb{R}_n = \{R_1, \ldots, R_n\}$ and its subsets $\mathbb{R}_k$ of size $k \geq 3$, three responses $R_x, R_y, R_z \in \mathbb{R}_k$ satisfy one of the following transitivity violations:

- **Circular inconsistency**: $C(R_x, R_y) = 1 \wedge C(R_y, R_z) = 1 \wedge C(R_z, R_x) \neq -1$. (2) (forming a preference cycle $R_x \succ R_y \succ R_z \not\prec R_x$)

- **Inequality inconsistency**: $C(R_x, R_y) = 0 \wedge C(R_y, R_z) = 0 \wedge C(R_x, R_z) \neq 0$. (3) (violating transitivity of indifference)

The **Non-Transitivity Ratio** is defined as $\text{NTR}_k = \frac{V_k}{\binom{n}{k}}$, where $V_k$ denotes the number of $k$-size subsets exhibiting either inconsistency type and $\binom{n}{k}$ represents the binomial coefficient counting all possible $k$-size subsets from $n$ elements.

### 2.2 TRUSTJUDGE

As shown in Algorithm 1, the TrustJudge framework is a probabilistic evaluation approach that preserves judgment entropy while resolving score-comparison conflicts and transitivity violations. The framework operates differently for single-score evaluation and pairwise comparison tasks, maintaining consistency between these two evaluation protocols.

For single-score evaluation, TrustJudge employs a *distribution-sensitive scoring* mechanism. Given a response $R$ to be assessed, the framework first prompts the LLM to score on a more fine-grained scale than original (e.g., a 100-point scale when the original scale is 5-point). It then transforms the resulting

---

**Algorithm 1** TrustJudge Evaluation Framework

---

**Require:** Responses $R_x$, $R_y$ (pairwise) or $R$ (single), expanded scores $\Theta'$ (range $[s'_{\min}, s'_{\max}]$), original range $[s_{\min}, s_{\max}]$
**Ensure:** Score $S$ or comparison $C(R_x, R_y)$
  1: **if** Single-Score Evaluation **then**
  2:  $P(s'_j|R) \leftarrow \mathrm{Softmax}(P_o(s'_j|R))$ {Normalize probabilities}
  3:  $S' \leftarrow \sum_{j=s'_{min}}^{s'_{max}} s'_j P(s'_j|R)$ {Expected expanded score}
  4:  $S \leftarrow S' \times \frac{s_{\max}-s_{\min}}{s'_{\max}-s'_{\min}}$ {Scale to original range}
  5:  **return** $S$
  6: **else** {Pairwise Comparison}
  7:  **Option A: PPL-Based**
  8:  $\mathrm{PPL}_1 \leftarrow \mathrm{PPL}(\mathcal{M}, R_x, R_y)$ {Perplexity of $R_x$ followed by $R_y$}
  9:  $\mathrm{PPL}_2 \leftarrow \mathrm{PPL}(\mathcal{M}, R_y, R_x)$ {Perplexity of reverse ordering}
 10:  $C(R_x, R_y) \leftarrow \begin{cases} C_{\mathrm{order}_1} & \text{if } \mathrm{PPL}_1 < \mathrm{PPL}_2 \\ C_{\mathrm{order}_2} & \text{otherwise} \end{cases}$ {Determine by comparing PPL}
 11:  **Option B: Likelihood-aware Aggregation**
 12:  $\mathbf{p}_1 \leftarrow \mathrm{Prob}(\mathcal{M}, R_x, R_y)$ {Probabilities for $R_x$ vs $R_y$}
 13:  $\mathbf{p}_2 \leftarrow \mathrm{Prob}(\mathcal{M}, R_y, R_x)$ {Probabilities for reverse comparison}
 14:  $m[k] \leftarrow \mathbf{p}_1[k] + \mathbf{p}_2[-k]$ for $k \in \{1, -1, 0\}$ {Aggregate both directions}
 15:  **return** $\arg\max_k m[k]$ {Select most probable outcome}
 16: **end if**

---

discrete probability distribution $P_o(s'_j \mid R)$ over the expanded score set $\Theta' = \{s'_{min}, \ldots, s'_{max}\}$ into logits $\ell_j$ for each candidate score $s'_j$. These logits are then processed by a softmax function which normalize the logits into a valid probability distribution $P(s'_j|R)$. Unlike other approaches such as G-Eval Liu et al. (2023), whose generated probabilities can violate $\sum_j P(s'_j \mid R) = 1$ because non-score tokens also influence the output, our method ensures a properly normalized distribution. The final score $S$ is computed as the expected value, scaled back to the original range $[s_{\min}, s_{\max}]$:

$$S = \left( \sum_{j=s'_{min}}^{s'_{max}} s'_j \cdot \frac{\exp(P_o(s'_j|R))}{\sum \exp(P_o(s'_k|R))} \right) \times \frac{s_{\max} - s_{\min}}{s'_{\max} - s'_{\min}}, \qquad (4)$$

where $P(s'_j|R)$ represents the original probability for score $s'_j$. This approach preserves the full entropy of the judge's assessment while producing continuous scores that maintain fine-grained distinctions between response qualities.

For pairwise comparison tasks, TrustJudge offers *likelihood-aware aggregation* methods to resolve transitivity inconsistencies. The first approach (Option A) leverages *perplexity-based (PPL-based) method* to break ties when the judge exhibits ambiguity. Given two responses $R_x$ and $R_y$, the framework computes the perplexity of both possible orderings ($R_x$ followed by $R_y$ and vice versa) under the judge model $\mathcal{M}$. The comparison result $C(R_x, R_y)$ is determined by selecting the ordering with lower perplexity:

$$C(R_x, R_y) = \begin{cases} C_{\mathrm{order}_1}, & \text{if } \mathrm{PPL}(\mathcal{M}, R_x, R_y) < \mathrm{PPL}(\mathcal{M}, R_y, R_x), \\ C_{\mathrm{order}_2}, & \text{otherwise.} \end{cases} \qquad (5)$$

The second approach (Option B) employs a bidirectional probability based method that combines preference probabilities from both orderings to reduce position bias. For each possible outcome $k \in \{1, -1, 0\}$ (representing $R_x$ preferred, $R_y$ preferred, or tie), the framework aggregates the probabilities from both orderings:

$$m[k] = \mathbf{p}_{\mathrm{order}_1}[k] + \mathbf{p}_{\mathrm{order}_2}[-k]. \qquad (6)$$

where $\mathbf{p}_{\text{order}_1}$ and $\mathbf{p}_{\text{order}_2}$ are the probability vectors for the two orderings. The final comparison result is determined by selecting the outcome with maximum aggregated probability $k^* = \arg\max_k m[k]$. This probabilistic approach significantly reduces circular and inequality transitivity violations while maintaining the scalability of pairwise comparisons.

By producing nearly continuous score distributions, a probabilistic judge makes exact equality between two responses much less likely than traditional discrete grading. To relax the tie criterion, we can introduce a tolerance hyper-parameter $\delta \geq 0$. Whenever the discrepancy between two responses—absolute score difference, PPL gap, or probability margin—does not exceed $\delta$, the pair is declared a tie, allowing users to tune the granularity of the final ranking without retraining the model. Although $\delta$ is set to 0 by default, we have conducted a thorough hyper-parameter sweep that confirms TrustJudge's reliability across a range of $\delta$ values; the results recommend a small positive $\delta$, because—even with $\delta = 0$—the framework already produces a noticeable number of ties.

## 3  THEORETICAL ANALYSIS

In this section, we formalize the theoretical weaknesses of current LLM-as-a-judge frameworks and prove how TrustJudge's components address them. The detailed analysis and derivation can be found at Appendix H. We start by proving that discrete scoring systems suffer from information loss.

**Theorem 3.1** (Information Loss of Discrete Scoring and Preservation in Expectation). *Let $p_{R_1}$ and $p_{R_2}$ be two distinct probability distributions over the score set $\Theta$ representing the judge model's assessment of two different responses, $R_1$ and $R_2$ (i.e., $p_{R_1} \neq p_{R_2}$). The discrete scoring function $f_{Discrete}$ can fail to distinguish between these two assessments, whereas the distribution-sensitive scoring function $f_{DS}$ provides a mechanism for their discrimination. Specifically:*

1. *(**Information Loss**): There exist $p_{R_1} \neq p_{R_2}$ with different conditional entropies, $H(S|R_1) \neq H(S|R_2)$, such that their discrete scores are identical: $f_{Discrete}(p_{R_1}) = f_{Discrete}(p_{R_2})$.*

2. *(**Information Preservation**): For the same distributions $p_{R_1}$ and $p_{R_2}$ constructed in (1), their distribution-sensitive scores are distinct: $f_{DS}(p_{R_1}) \neq f_{DS}(p_{R_2})$.*

For pairwise comparisons [2], we formalize how the PPL-based method reduces the uncertainty caused by ambiguity, the proof of which is deferred to the Appendix H.

**Proposition 3.2** (Uncertainty Reduction via PPL-based Method). *Let $H(C|\pi)$ be the Shannon entropy of the judge model's outcome distribution. In an **ambiguous regime**, this entropy is maximized, $H(C|\pi) \approx \log|\mathcal{C}|$. We define a confidence distribution $p_{conf}$ based on the perplexity of the rationale $J_k$ for each outcome $k$:*

$$p_{conf}(k) \propto \exp(-\gamma \cdot PPL(J_k)), \quad \text{for a scaling constant } \gamma > 0. \tag{7}$$

*If the rationale perplexities are not all equal, then $p_{conf}$ is non-uniform. By the properties of Shannon entropy, this implies its entropy is strictly less than the maximum:*

$$H(p_{conf}) < \log|\mathcal{C}|. \tag{8}$$

*Thus, the PPL-based method makes its decision using a more certain (lower-entropy) signal.*

## 4  EXPERIMENTS

**Setup**  Our dataset combines both the 80 questions from MT-Bench Zheng et al. (2023) and the 500 challenges from ArenaHard Li et al. (2024). MT-Bench provides broad coverage of diverse instructions across eight categories including writing, roleplay, and reasoning, while ArenaHard offers particularly challenging queries drawn from real-world user interactions. For each question, we sample candidate responses from diverse LLMs. Under the single-comparison inconsistency protocol, we construct a dataset of 10.8k instances, where each instance corresponds to a pair of responses annotated with their single scores and the induced pairwise preference. Under the pairwise transitivity inconsistency protocol, we collect 43.2k pairwise relations for $k = 4$ and 50.4k for $k = 5$, each derived from the corresponding $k$-response sets to evaluate transitivity. For each question,

---

[2] Please see more analysis of the bidirectional probability based method in Appendix H.2

we collected responses from a diverse set of large language models with varying capabilities. All gold-standard scores and pairwise comparison results for these responses are verified through human review. The final dataset is carefully balanced, ensuring uniform score distributions across every rating level for both single-score and pairwise-comparison scenarios. The detailed model information and inference hyperparameters used in our test data creation are listed in Appendix D, and the detailed category distribution across our datasets is provided in Appendix G. Beyond the core framework, we also extend our approach to multi-dimensional evaluation, as detailed in Appendix F.

We evaluate both the inconsistencies and accuracies. For inconsistencies, we use: (1) the **Conflict Ratio** (CR, Definition 2.1) and (2) the **Non-Transitivity Ratio** (NTR, Definition 2.2). For accuracies, we employ (1) **Win Rate** for single-score evaluation: the fraction of instances whose score sides with the reference, which highlights protocol differences more sharply than MSE or MAE. (2) **Exact Match** for pairwise comparison: given the ground-truth order $A \succ B$, only a verbatim output of $A \succ B$ counts; any reversal or tie is wrong—an all-or-nothing metric. Parameter $k$ (Def. 2.2) governs the subset size for pairwise transitivity checks. Larger $k$ captures higher-order cycles at cost $\binom{n}{k}$; $k = 3$ yields too few triples to discriminate models, so we report $k = 4, 5$.

**Baselines** We establish two fundamental baseline approaches for comparison with TrustJudge. For single-score evaluation, we implement: (1) the standard raw scoring method that directly outputs absolute scores (1-5 scale), as used in MT-Bench; and (2) G-Eval-style probability summation that calculates the total probability mass across possible scores without applying softmax normalization. Following previous work Wang et al., **pairwise baseline mitigates position bias by evaluating each response pair twice (reversed order) and record differing outcomes as ties.** All baselines use the identical judge model and the same detailed prompt (see Appendix C) as TrustJudge, enabling direct comparison of inconsistency metrics (CR and NTR) and accuracy metrics (Win Rate and Exact Match) under identical conditions.

**Evaluated LLMs** Our experiments comprehensively evaluate TrustJudge across a diverse set of popular LLMs, covering both open-source and proprietary families with varying parameter sizes. Specifically, we include: **Llama-3.2-3B**, **Llama-3.1-8B**, and **Llama-3.1-70B** Grattafiori et al. (2024), three instruction-tuned models from the Llama-3 series, representing small, medium, and large-scale open-source LLMs, respectively. We also evaluate **GPT-3.5-Turbo** OpenAI (2023) and **GPT-4o** OpenAI (2024), two widely-used proprietary models from OpenAI, included for their strong performance in both general and evaluation-specific benchmarks. Additionally, we assess the **Qwen2.5-7B**, **Qwen2.5-14B**, **Qwen2.5-32B** Yang et al. (2024), **Gemma-2-2b**, **Gemma-2-9B**, and **Gemma-2-27B** Riviere et al. (2024) to demonstrate TrustJudge's generalization across model types and sizes. For all evaluations, we use the instruction-tuned or SFT variants of each model, consistent with their intended use as judge LLMs. All judge models are applied with identical evaluation prompts and configurations to ensure fair comparison.

**Main Results** The experimental results comparing TrustJudge with baseline approaches across multiple model families and sizes are summarized in Table 1 and Figure 2. The key findings are:

Table 1: Results for two experiments: (1) Score-Comparison Inconsistency (CR) comparing raw-score baseline, G-Eval probability-summation, and TrustJudge's distribution-sensitive scoring; (2) Pairwise Transitivity Inconsistency (NTR$_{k=4,5}$) comparing two-pass swap-order baseline versus TrustJudge's likelihood-aware aggregation. Win rate quantifies scoring precision by measuring the proportion of test instances where a method's score is nearest the ground truth with results presented on both 5-point and 100-point scales. Exact match quantifies comparison consistency by measuring the proportion of pairwise method outcomes that perfectly align with dataset annotations.

| Model | CR (%) | | | NTR$_{k=4}$ (%) | | NTR$_{k=5}$ (%) | | Ours vs Baseline | | Ours vs G-Eval | | Pairwise Exact Match | |
|---|---|---|---|---|---|---|---|---|---|---|---|---|---|
| | Baseline | G-Eval | Ours | Baseline | Ours | Baseline | Ours | 5-scale | 100-scale | 5-scale | 100-scale | Baseline | TrustJudge |
| Llama-3.2-3B-Instruct | 36.65 | 29.50 | **29.15** | 32.42 | **8.07** | 54.69 | **17.76** | 45.41 | **54.66** | 62.21 | **51.03** | 72.06 | **78.91** |
| Llama-3.1-8B-Instruct | 29.73 | 25.31 | **23.75** | 20.26 | **3.79** | 37.03 | **8.46** | 56.84 | **51.88** | 59.61 | **51.24** | 75.67 | **81.68** |
| Llama-3.1-70B-Instruct | 23.32 | 15.77 | **14.89** | 7.23 | **1.94** | 15.22 | **4.40** | 51.77 | **54.53** | 64.22 | **55.27** | 80.42 | **81.61** |
| GPT-4o | 27.95 | 23.18 | **22.60** | 11.70 | **2.83** | 24.33 | **6.01** | 50.31 | **55.60** | 65.11 | **53.43** | 78.67 | **81.51** |

**TrustJudge significantly reduces evaluation inconsistencies across all model sizes.** Our experiments demonstrate that TrustJudge achieves superior consistency compared to both direct scoring

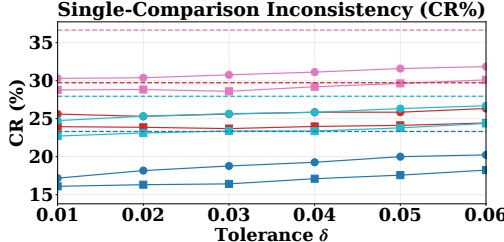

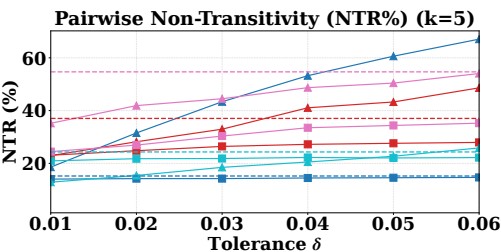

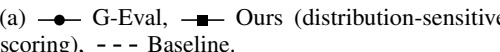

(a) —●— G-Eval, —■— Ours (distribution-sensitive scoring), --- Baseline.

(b) —▲— Ours (PPL-based method), —■— Ours (likelihood-aware aggregation), --- Baseline.

Figure 2: Results for single-comparison inconsistency (left) and pairwise transitivity inconsistency (right) across tolerance $\delta$: Llama-3.1-70B (blue), Llama-3.1-8B (red), Llama-3.2-3B (pink), and GPT-4o (green). Colors correspond to different judge models, while markers distinguish evaluation methods as described in each subfigure. For single-score in the experiment on the left, $\delta$ is a tolerance proportion on the original rating scale; For pairwise (PPL-based) in the experiment on the right, $\delta$ is the threshold on the difference in perplexity between the two presentation orders; For pairwise (Likelihood-aware aggregation) in the experiment on the right, $\delta$ is the threshold on the confidence gap between the top two aggregated outcomes.

baselines and G-Eval approaches. The proposed method achieves substantial reductions in Conflict Ratios, delivering absolute improvements of 4.78%–8.43% over the baseline approaches. Moreover, Trustjudge consistently surpasses G-eval by approximately 1–2% across every experimental setting. More importantly, TrustJudge dramatically lowers transitivity violations in pairwise comparisons, with $\text{NTR}_{k=5}$ violations reduced by 10.82%-36.93% absolute. For instance, Llama-3.2-3B shows the most substantial improvement, decreasing $\text{NTR}_{k=5}$ from 54.69% to just 17.76% with TrustJudge. These consistency improvements are particularly notable because they are achieved without requiring additional training or fine-tuning of the base models.

**TrustJudge maintains and often improves evaluation accuracy while reducing inconsistencies.** TrustJudge demonstrates that both consistency and accuracy can be achieved simultaneously. Trust-Judge improves exact match rates by 1.19%-6.85% across different model sizes compared to baseline approaches, with the most significant gains observed for smaller models (6.85% improvement for Llama-3.2-3B). In pairwise evaluations, TrustJudge achieves win rates of 45.41%-65.11% against both baseline methods and G-Eval approaches. The method performs particularly well on fine-grained 100-point scoring and maintains strong performance on 5-point scales. This accuracy preservation is crucial for practical applications where both reliable and precise evaluations are required.

**TrustJudge exhibits robust tolerance-aware gains across judge families and evaluation protocols.** A fine-tuned tolerance ($\delta$) often yields superior outcomes, as a smaller tolerance reduces ambiguity. Conversely, a larger tolerance introduces greater uncertainty. It's important to note that even with a tolerance of zero, ties can still occur. The TrustJudge scoring and aggregation method effectively mitigates inconsistencies. Notably, its benefits are evident across various tolerance settings, demonstrating its robustness and effectiveness.

Table 2: Ablation study where "L" refers to LLaMA and "G" to GPT. Single Score Components report CR and Pairwise Comparison Components report $\text{NTR}_{k=4}$.

| Components | L-3.1-8B | L-3.1-70B | G-3.5-Turbo | G-4o |
|---|---|---|---|---|
| *Single Score Components* | | | | |
| 5-scale Baseline | 29.73 | 23.32 | 24.35 | 27.95 |
| + Softmax | 26.10 | **17.08** | 24.03 | 25.50 |
| + 100-scale | **24.54** | 17.94 | **22.10** | **24.01** |
| *Pairwise Comparison Components* | | | | |
| Baseline | 20.26 | 7.23 | 14.01 | 11.70 |
| + Likelihood | **3.79** | **1.94** | 6.26 | **2.83** |
| + PPL-Based | 6.56 | 2.18 | **4.80** | 4.48 |

**Ablation Study** To evaluate the contribution of different components in TrustJudge, we conduct an ablation study by systematically removing key elements: (1) the softmax normalization, (2) the 100-point granularity enhancement for single-score evaluation, and (3) the pairwise comparison strategies (likelihood-aware aggregation and PPL-based methods). We also examine performance variations across different judge LLMs to demonstrate TrustJudge's model-agnostic properties.

Table 2 reveals several key findings. For single score components, the 5-scale baseline shows the highest inconsistency rates across all models (39.73% for Llama-3.1-8B, 27.5% for GPT-4o), indicating the importance of TrustJudge's enhancements. Adding softmax normalization reduces inconsistency by 0.32%-6.24% absolute across models, while incorporating 100-scale granularity yields improvements (up to 5.19% reduction from 5-scale).

In pairwise comparison components, the baseline shows moderate performance (20.26% inconsistency for Llama-3.1-8B). The likelihood-aware aggregation strategy achieves the best results overall, reducing inconsistency to as low as 1.94% for Llama-3.1-70B and 2.83% for GPT-4o. The PPL-based comparison shows substantial gains over baseline (16.47% absolute improvement for Llama-3.1-8B) while offering practical advantages in implementation, as it operates directly on sequence probabilities without requiring explicit win/tie/lose position identification .

The consistent performance patterns across model architectures (from 8B to 70B parameters) demonstrate that TrustJudge's benefits are not model-specific but derive from its methodological innovations. Larger models generally achieve better absolute performance, with Llama-3.1-70B and GPT-4o showing particularly strong results when using TrustJudge.

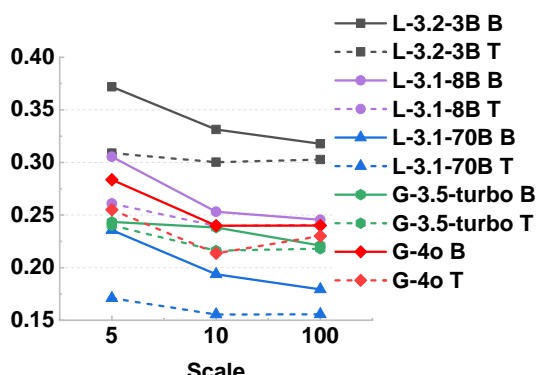

Figure 3: Effect of scoring granularity on *Conflict Ratio* across judge models. We measure the Conflict Ratio (CR) under three scoring scales: 5-point, 10-point, and 100-point. "L" refers to LLaMA models and "G" to GPT models; "B" denotes baseline scoring, while "T" represents TrustJudge.

**Increasing Score granularity reduces inconsistency.** As shown in Figure 3, increasing the scoring scale from 5 to 100 points consistently reduces the Conflict Ratios. Furthermore, TrustJudge (T) achieves lower inconsistency than the baseline (B) under all granularities, demonstrating its effectiveness in preserving scoring fidelity. The benefit is especially pronounced for larger models such as Llama-3.1-70B and GPT-4o.

**Generalization Experiment** To systematically validate TrustJudge's cross-architectural adaptability and practical value for alignment training, we evaluate the framework across 12 model variants spanning four major architectures (Qwen, Gemma, Llama, GPT) with various parameter sizes. The experiments cover both single-response scoring and pairwise comparison scenarios. Note that we set $k = 4$ for pairwise comparison.

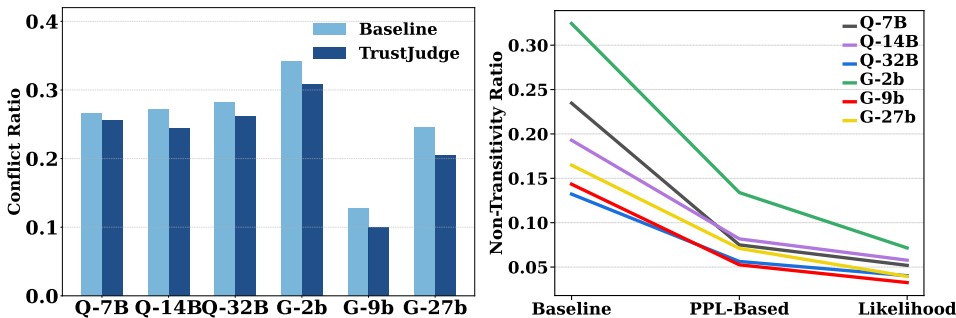

Figure 4: Performance of TrustJudge with LLMs of different Sizes and Structures. Note that Qwen-2.5 is denoted as Q and Gemma-2 as G.

Figure 4 demonstrates three key findings through comprehensive architectural comparisons: *Architecture-agnostic consistency improvement*. The distribution-sensitive scoring achieves consistent

reductions in single-instance conflict ratios across all tested architectures. Moreover, inconsistency varies markedly across architectures: Gemma consistently outperforms Qwen of comparable size.

*Transitivity violation reversal.* The proposed likelihood-aware aggregation strategy substantially mitigates non-transitivity patterns across model variants. Remarkably, this approach enables mid-sized models to surpass the transitivity performance of significantly larger baseline models under controlled evaluation settings.

*Size-performance decoupling.* While model capacity naturally correlates with lower inconsistency rates, TrustJudge effectively narrows the performance disparity between small and large models. This capability highlights the framework's potential to enhance the practical utility of resource-efficient models for alignment tasks. Notably, bigger is not always better: the 9B Gemma actually exhibits lower inconsistency than its 27B sibling.

Table 3: Performance of TrustJudge for Llama-3.1-8B and DeepSeek-R1-Distill-Llama-8B.

| Model | CR (%) | | | $\text{NTR}_{k=4}$ (%) | | | $\text{NTR}_{k=5}$ (%) | | |
|---|---|---|---|---|---|---|---|---|---|
| | Baseline | G-Eval | Ours | Baseline | Likelihood | PPL-Based | Baseline | Likelihood | PPL-Based |
| Llama-3.1-8B | 29.73 | 25.31 | **23.75** | 20.26 | **3.79** | 6.80 | 37.03 | **8.46** | 16.20 |
| DeepSeek-R1-Distill-Llama-8B | 58.75 | 53.63 | **49.28** | 44.61 | **11.43** | 25.16 | 63.98 | **18.50** | 41.78 |

**Reasoning model results**    As shown in Table 3, The reasoning model's significantly higher inconsistency rates suggest a potential catastrophic forgetting of judge capabilities due to reinforcement training on mathematical data Guo et al. (2025). This finding is noteworthy as it highlights the challenges that arise when models are trained on specialized tasks, such as mathematical reasoning, which can inadvertently lead to the degradation of their performance in other critical areas like judging. Despite this, TrustJudge remains effective in improving judge performance, demonstrating its robustness and adaptability in enhancing the model's capabilities across different domains.

**Using Trustjudge for Rewarding Models**    Table 4 shows TrustJudge's effect when used as a reward signal for GRPO. We run GRPO on Qwen2.5-7B-Instruct using a multi-task corpus with 8,600 training examples and 265 validation examples covering summarization, math reasoning, instruction following, and related tasks Feng et al. (2025). For evaluation, all three models are assessed by an external LLM judge (GPT-4o-mini) under both the Baseline

Table 4: GRPO results on Qwen2.5-7B with different reward designs.

| Model | Avg Reward (Baseline) | Avg Reward (TrustJudge) |
|---|---|---|
| Qwen2.5-7B-Instruct | 0.5521 | 0.6114 |
| Qwen2.5-7B-Instruct_baseline | 0.5377 | 0.5980 |
| Qwen2.5-7B-Instruct_TrustJudge | **0.5706** | **0.6347** |

and TrustJudge scoring protocols. The model trained with the TrustJudge reward achieves the best performance under both evaluation protocols, obtaining average rewards of 0.5706 under the Baseline protocol and 0.6347 under the TrustJudge protocol. In contrast, the model trained with the Baseline reward fails to improve over the original Qwen2.5-7B-Instruct and instead exhibits a decline in performance under both protocols. These results suggest that TrustJudge provides a more informative and robust reward signal, regardless of the downstream evaluation protocol. Additional training and evaluation details are provided in Appendix E.

The results establish TrustJudge's robust generalizability across: (1) different model families and scales, maintaining consistent inconsistency reduction regardless of architecture; and (2) diverse applications including direct evaluation and reward modeling for GRPO training. This versatility stems from TrustJudge's architecture-agnostic probabilistic design and fine-grained scoring approach.

## 5    CONCLUSION

We presented TrustJudge, a novel probabilistic evaluation framework designed to address fundamental inconsistencies in current LLM-as-a-judge paradigms. Through systematic analysis, we identified two critical issues: Score-Comparison Inconsistency due to information loss in discrete scoring systems, and Pairwise Transitivity Inconsistency stemming from ambiguous tie judgments. TrustJudge

introduces distribution-sensitive probabilistic scoring, preserving judgment entropy, and likelihood-aware aggregation strategies to effectively mitigate these inconsistencies.

Empirical results demonstrate that TrustJudge significantly reduces Score-Comparison inconsistency and Pairwise Transitivity inconsistency across various LLM architectures and scales. Crucially, these improvements do not compromise evaluation accuracy, achieving enhancements in exact match rates and win rates compared to established baselines. Our ablation and generalization studies confirm the robustness and model-agnostic applicability of TrustJudge. TrustJudge offers both theoretical insights and practical solutions for enhancing the reliability and credibility of automated LLM evaluations, contributing towards more trustworthy and consistent use of large language models in research and applications. We also discuss the limitations of our approach in Appendix B.

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

## A  RELATED WORK

**Traditional Discrete Evaluation Protocols**   LLM-as-a-judge frameworks have become widely adopted for their scalability and cost-efficiency in evaluating large language models. Early works predominantly relied on discrete evaluation protocols, including coarse single-score ratings and pairwise preference comparisons. MT-Bench and Chatbot Arena Zheng et al. (2023) demonstrated the feasibility of using powerful LLMs such as GPT-4 as judges, achieving high agreement with human preferences, while also noting issues such as verbosity and position bias. ArenaHard Li et al. (2024) proposed an automated benchmark construction pipeline and introduced Arena-Hard-Auto, a challenging benchmark curated without human-in-the-loop, which relies on LLMs to produce and evaluate responses.

AlpacaEval Dubois et al. (2024) highlighted persistent biases in LLM-based evaluation such as a preference for longer outputs. Their proposed length-controlled regression analysis mitigated this issue and improved correlation with human preferences. PandaLM Wang et al. introduced a pairwise judgment protocol that incorporates subjective dimensions like clarity, formality, and instruction adherence. Its judge model outperformed even GPT-4 in certain domains and was used to tune instruction-following models.

Other works such as Trivedi et al. (2024); Saha et al. (2023); Que et al. (2024); Ye et al. (2023); Fernandes et al. (2023); Bai et al. (2023); Wang et al. (2025a) developed various discrete evaluation techniques, including majority voting, scalar ratings, skill-wise decomposition, and output-based scoring. While these methods brought interpretability and practical value, they were still constrained by coarse-grained annotations and did not fully resolve contradictions between scoring types or internal inconsistencies. Notably, Fernandes et al. (2023) proposed AUTOMQM for machine translation, which incorporated structured error labeling but remained within the paradigm of fixed-score prompting. Bai et al. (2023) proposed a language model examiner framework combining scoring and ranking, but without entropy-aware mechanisms. Similarly, Wang et al. (2025a) employed output-based scoring for software engineering tasks, emphasizing alignment with human evaluation but without probabilistic modeling. Additionally, Trivedi et al. (2024) used iterative self-rationalization for enhancing model rationales but still within a discrete scoring.

**Probabilistic and Fine-Grained Evaluation Methods**   To overcome the limitations of discrete judgments, recent research has explored probabilistic evaluation strategies. G-EVAL Liu et al. (2023) introduced softmax-normalized score prediction over a fine-grained rating scale using chain-of-thought prompting and form-filling, improving alignment with human preferences. Wang et al. (2025b) further examined extracting fine-grained preferences by leveraging the distributional output of judge models, demonstrating that methods incorporating distributional judgments significantly outperform traditional greedy decoding across various evaluation scenarios.

**Instruction:**
Below is a question and a candidate response from a user . Please act as an impartial judge and evaluate the quality of the responses provided by two AI assistants to the user question displayed below. You should choose the assistant that follows the user's instructions and answers the user's question better. Your evaluation should consider factors such as the helpfulness, relevance, accuracy, depth, creativity, and level of detail of their responses. Begin your evaluation by comparing the two responses and provide a short explanation. Avoid any position biases and ensure that the order in which the responses were presented does not influence your decision. Do not allow the length of the responses to influence your evaluation. Do not favor certain names of the assistants. Be as objective as possible. After providing your brief explanation, you *must* output the final verdict by strictly following this format: \"[A]\" if assistant A is better, \"[B]\" if assistant B is better, and \"[C]\" for a tie. Provide a result exclusively in **square brackets** (e.g., Verdict: [A]).

**Question:**
Compose an engaging travel blog post about a recent trip to Hawaii, highlighting cultural experiences and must-see attractions.

**Answer A:**
Clouds shrouded Luau nights at various hotels **(Did you know you can also find luau options on the beach?)** are basically more turn-key opportunities to meet like-minded travelers at the resort complexes. A quirky day I revisited that place just made my visit almost two weeks and nine days based on prior reviews I had. Probably either searching the night markets or actually hopping in the ocean in the afternoon somewhere that first island had a lot of gratifying moments of vibrancy. Due to mixed reviews from prior travelers, shrimp tempura happened last thing on the last night in one restaurant.

**Answer B:**
Aloha from paradise! My recent trip to Hawaii was a whirlwind of vibrant culture, breathtaking landscapes, and unforgettable experiences. Immersing myself in the local way of life was truly enriching. I discovered the importance of respecting the \u0101ina (land) and the deep connection Hawaiians have to their heritage. Learning a few basic Hawaiian words and phrases helped me connect with the locals and gain a deeper appreciation for their culture. Of course, I also explored some iconic landmarks. The natural beauty of the islands is simply awe-inspiring, from the cascading waterfalls to the pristine beaches.

**Judgement:**
Assistant B provides a much better response to the user's question. Assistant B creates an engaging blog post that highlights cultural experiences in Hawaii, such as respecting the land, connecting with locals, and learning about Hawaiian heritage. The assistant also mentions exploring iconic landmarks like waterfalls and beaches. On the other hand, Assistant A's response is disjointed and lacks coherence, jumping between random experiences without a clear focus on cultural aspects or must-see attractions. Overall, Assistant B's response is more helpful, relevant, accurate, and detailed in addressing the user's request for an engaging travel blog post about Hawaii.\n\n**Verdict: [B]**

{"B": 0.63743090092823145, "A": 0.24067150787693196, "C": 0.12532604394079833…}

Figure 5: Example of a pairwise evaluation prompt.

Our work builds upon and extends this direction by proposing TrustJudge, a probabilistic evaluation framework that preserves judgment entropy and explicitly resolves both score-comparison and pairwise transitivity inconsistencies in LLM-as-a-judge paradigms.

## B    LIMITATIONS

Despite the demonstrated efficacy of TrustJudge, our approach still has some inherent limitations. Firstly, the performance of TrustJudge is fundamentally dependent on the instruction-following capabilities of the employed evaluation models. Smaller-scale language models often exhibit weaker instruction comprehension and execution capabilities, which could result in failure to yield valid scores or comparisons. Consequently, the quality and reliability of TrustJudge evaluations are directly tied to the underlying judge model's competence, emphasizing the importance of model ability.

## C    PROMPT EXAMPLES

The following figures 5 and 6 provide examples of evaluation prompts used to assess responses. The first figure shows a pairwise comparison prompt, where two responses are compared and one is selected as better. The second figure illustrates a single-score evaluation prompt with the 5-point scale, where a response is rated based on quality metrics such as helpfulness and relevance. These examples are intended to support clarity and consistency in LLM-as-a-judge evaluation tasks.

**Instruction:**
Please act as an impartial judge and carefully evaluate the quality of the AI assistant's response to the user question provided below. Your evaluation should consider factors such as the helpfulness, relevance, accuracy, depth, creativity, and level of detail of the response. Provide a concise, objective explanation summarizing your evaluation. IMPORTANT: After your explanation, give only a single, final numeric score on a scale of 1 to 5 in **square brackets**, strictly adhering to the following format: \"Score: [X]\", where X is your numeric score (e.g., \"Score: [2]\").

**Question:**
Compose an engaging travel blog post about a recent trip to Hawaii, highlighting cultural experiences and must-see attractions.

**Answer:**
I'm glad you enjoyed your recent trip to Hawaii! In this travel blog post, I will share with you the cultural experiences and must-see attractions that made your journey unforgettable. Get ready to immerse yourself in the beauty and richness of Hawaiian culture as we explore vibrant luaus, traditional hula performances, and delicious local cuisine. Don't forget to check out iconic attractions like the stunning beaches of Waikiki, the volcanic wonders of Hawaii Volcanoes National Park, and the historical significance of Pearl Harbor. So grab your sunscreen, put on your leis, and let's embark on a virtual journey to the tropical paradise of Hawaii!

**Judgement:**
The response is friendly and engaging, directly addressing the user's request for an engaging travel blog post about a recent trip to Hawaii. It includes highlights of cultural experiences and must-see attractions, painting a vivid picture for the reader. The use of descriptive language and inviting tone creates an immersive experience for the audience, making them eager to explore Hawaii through the blog post. The response fulfills the user's request effectively and offers a comprehensive overview of the trip, showcasing the cultural richness and key attractions of Hawaii.\n\n**Score: [4]**

{"4": 0.3775228458971041,
"3": 0.6224299462058022,
"2": 4.659003155833513e-05,
"5": 5.175684992981448e-07,
...}

Figure 6: Example of a single-score evaluation prompt with the 5-point scale.

## D  INFERENCE SETTINGS

Specifically, we included strong open-source models such as Llama-3-Athene-70B Frick et al. (2024), Llama-3-70B-Instruct Grattafiori et al. (2024), and Llama-3-8B-Instruct Grattafiori et al. (2024); strong closed-source models such as GPT-4o OpenAI (2024), GPT-4-Turbo OpenAI (2024), and Claude 3 Sonnet Anthropic (2024); weak open-source models including WizardLM-13B-v1.2 Xu et al. (2023); WizardLM Team (2023), Vicuna-7B Chiang et al. (2023); LMSYS (2023), and Alpaca-13B Chavinlo (2023); and weak closed-source models such as Claude 3 Haiku Anthropic (2024) and GPT-3.5-Turbo OpenAI (2023). We use batched inference of vLLM to accelerate the generation and judging process, setting the temperature to 1.0, the maximum number of tokens to 2048, and providing the top 20 log probabilities for each generated token.

## E  GRPO TRAINING SETUP

We conduct GRPO training on Qwen2.5-7B-Instruct using the multi-task corpus introduced in Feng et al. (2025), following the 8,600/265 training-validation split. The Baseline and TrustJudge conditions are trained under an identical GRPO configuration, with rollout groups of size 8, a batch size of 256, a maximum prompt length of 9000 tokens, and a maximum response length of 1024 tokens. The actor and critic are optimized with Adam using learning rates of $1 \times 10^{-6}$ and $1 \times 10^{-5}$, respectively, for 150 GRPO steps. Unless otherwise noted, all other optimization hyperparameters follow the same settings across the two runs. The only difference lies in the reward function: one uses the Baseline scalarized 0–9 score, whereas the other uses the TrustJudge-based reward.

Figure 7 shows the training reward trajectories of Qwen2.5-7B-Instruct during GRPO under the shared hyperparameter setting, comparing the Baseline and TrustJudge reward formulations. Both runs exhibit steadily increasing rewards over the course of training. However, the TrustJudge-based run achieves consistently higher reward than the Baseline-based run across most of the 150 GRPO steps, suggesting that TrustJudge provides a more informative learning signal for policy optimization.

Figure 8 reports per-task validation rewards assigned by an external LLM judge (GPT-4o-mini) throughout GRPO training, with greedy decoding used at evaluation time. For each dataset in the multi-task mixture, including instruction following, long-document summarization, GSM8K, essay tasks, English–French translation, and WildChat, solid curves denote Qwen2.5-7B-Instruct trained with the TrustJudge reward, whereas dashed curves denote the counterpart trained with the Baseline reward. Across nearly all tasks, TrustJudge matches or outperforms the Baseline reward. This pattern is consistent with the aggregate improvements reported in Table 4, suggesting that the benefits of TrustJudge are broadly distributed across domains rather than driven by gains on any single dataset.

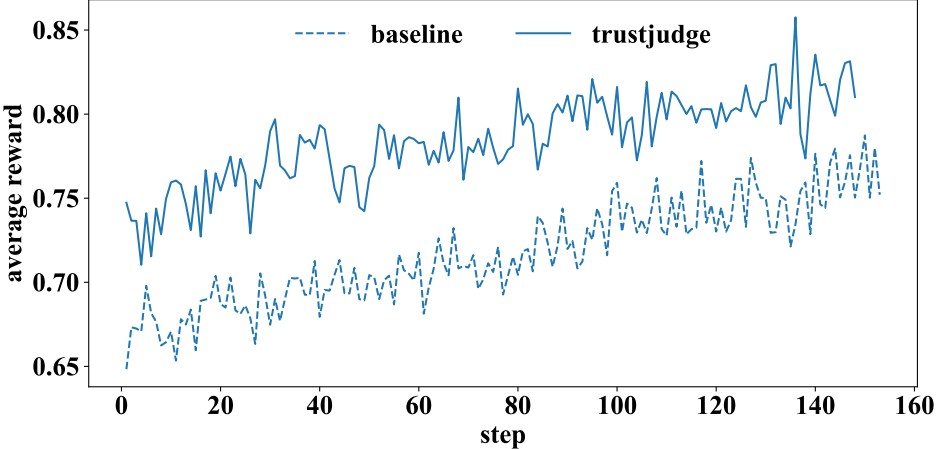

Figure 7: Training reward during GRPO training of Qwen2.5-7B-Instruct with Baseline and Trust-Judge rewards.

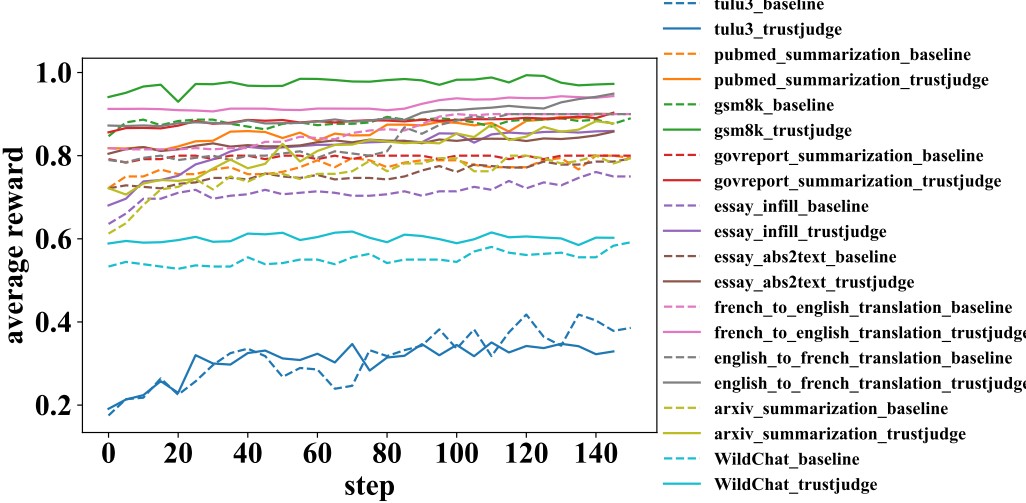

Figure 8: Per-task validation reward across GRPO training steps under an external LLM judge. Solid lines correspond to Qwen2.5-7B-Instruct trained with the TrustJudge reward and dashed lines correspond to Qwen2.5-7B-Instruct trained with the Baseline reward.

## F    EXTENSION TO MULTI-DIMENSIONAL EVALUATION

**Setup.**    To assess whether TrustJudge can be extended to multi-dimensional evaluation, we evaluate three sub-dimensions: factuality, coherence, and helpfulness. We randomly sample 120 questions from Arena Hard dataset. For each question, candidate responses and judgements are generated by

models from the Llama, Qwen, Gemma, and GPT families. Each sub-dimension uses a dimension-specific prompt, which we show in Figures 9 and 10 respectively.

For each sub-dimension, we independently compute two degrees of inconsistencies: (i) Score–Comparison Inconsistency reported as $CR$ and Pairwise Transitivity Inconsistency reported as $NTR_k$ for $k \in \{3, 4\}$. For brevity, Table 5 present the averages of these metrics across the three sub-dimensions, while all metrics are computed per dimension as specified above.

---

**Instruction for Factuality:**

Please act as an impartial judge and carefully evaluate the \*\*factuality\*\* of the AI assistant's response to the user question provided below. Your evaluation should consider factors such as the correctness of the information, the verifiability of claims, the accuracy of data and references, and the absence of factual errors or misleading statements. Provide a concise, objective explanation summarizing your evaluation. IMPORTANT: After your explanation, give only a single, final numeric score on a scale of 1 to 5 in \*\*square brackets\*\*, strictly adhering to the following format: \"Score: [X]\", where X is your numeric score (e.g., \"Score: [2]\").

**Instruction for Coherence:**

Please act as an impartial judge and carefully evaluate the \*\*coherence\*\* of the AI assistant's response to the user question provided below. Your evaluation should consider factors such as logical flow, clarity of structure, consistency of argumentation, appropriate organization of ideas, and overall readability. Provide a concise, objective explanation summarizing your evaluation. IMPORTANT: After your explanation, give only a single, final numeric score on a scale of 1 to 5 in \*\*square brackets\*\*, strictly adhering to the following format: \"Score: [X]\", where X is your numeric score (e.g., \"Score: [2]\").

**Instruction for Helpfulness:**

Please act as an impartial judge and carefully evaluate the \*\*helpfulness\*\* of the AI assistant's response to the user question provided below. Your evaluation should consider factors such as relevance to the user's needs, depth of insight, practical utility, clarity of guidance, and effectiveness in addressing the question. Provide a concise, objective explanation summarizing your evaluation. IMPORTANT: After your explanation, give only a single, final numeric score on a scale of 1 to 5 in \*\*square brackets\*\*, strictly adhering to the following format: \"Score: [X]\", where X is your numeric score (e.g., \"Score: [2]\").

Figure 9: Example of single-score prompts for multi-dimension evaluation.

---

**Instruction for Factuality:**

Below is a question and two candidate responses from AI assistants to the user question displayed below. Please act as an impartial judge and evaluate which response is more factually accurate. Your evaluation should consider factors such as the correctness of information, the verifiability of claims, the accuracy of data and references, and the absence of factual errors or misleading statements. Begin your evaluation by comparing the two responses and provide a concise explanation of which is more factually sound. Avoid any position biases and ensure that the order of the responses, their length, or the names of the assistants do not influence your decision. After providing your explanation, output only the final verdict by strictly following this format: \"[A]\" if assistant A's response is more factually accurate, \"[B]\" if assistant B's response is more factually accurate, and \"[C]\" for a tie, exclusively in square brackets (e.g., Verdict: [A])

**Instruction for Coherence:**

Below is a question and two candidate responses from AI assistants to the user question displayed below. Please act as an impartial judge and evaluate which response is more coherent. Your evaluation should consider factors such as logical flow, clarity of structure, consistency of argumentation, appropriate organization of ideas, and overall readability. Begin your evaluation by comparing the two responses and provide a concise explanation of which is more coherent. Avoid any position biases and ensure that the order of the responses, their length, or the names of the assistants do not influence your decision. After providing your explanation, output only the final verdict by strictly following this format: \"[A]\" if assistant A's response is more coherent, \"[B]\" if assistant B's response is more coherent, and \"[C]\" for a tie, exclusively in square brackets (e.g., Verdict: [A]).

**Instruction for Helpfulness :**

Below is a question and two candidate responses from AI assistants to the user question displayed below. Please act as an impartial judge and evaluate which response is more helpful. Your evaluation should consider factors such as relevance to the user's needs, depth of insight, practical utility, clarity of guidance, and effectiveness in addressing the question. Begin your evaluation by comparing the two responses and provide a concise explanation of which is more helpful. Avoid any position biases and ensure that the order of the responses, their length, or the names of the assistants do not influence your decision. After providing your explanation, output only the final verdict by strictly following this format: \"[A]\" if assistant A's response is more helpful, \"[B]\" if assistant B's response is more helpful, and \"[C]\" for a tie, exclusively in square brackets (e.g., Verdict: [A]).

Figure 10: Example of pairwise prompts for multi-dimension evaluation.

---

**Results and analysis.** Extending the judge to three axes—factuality, coherence, and helpfulness—yields a clear reduction in inconsistency. With dimension-specific prompts and per-dimension computation, we observe drops on every model and on both of the metrics: CR decreases by roughly 5.13%–11.03%, while $NTR_3$ and $NTR_4$ fall more sharply, on average by 11.23%–24.99%.

Table 5: Results for multi-dimensional evaluation across three sub-dimensions—factuality, coherence, and helpfulness. For each sub-dimension, CR and $\text{NTR}_k$ are computed independently; tables report the mean across sub-dimensions.

| Model | CR (%) | | $\text{NTR}_{k=3}$ (%) | | $\text{NTR}_{k=4}$ (%) | |
|---|---|---|---|---|---|---|
| | Baseline | Ours | Baseline | Ours | Baseline | Ours |
| Gemma-2-27b-it | 49.43 | **44.30** | 19.60 | **8.20** | 48.76 | **22.41** |
| Qwen2.5-32B-Instruct | 45.73 | **37.87** | 17.38 | **7.89** | 42.55 | **22.36** |
| Llama-3.1-70B-Instruct | 52.20 | **41.47** | 18.29 | **5.48** | 44.65 | **16.21** |

The pattern is most visible with Llama-3.1-70B, where $\text{NTR}_4$ contracts from 44.65% to 16.21% and $\text{NTR}_3$ from 18.29% to 5.48%, alongside a CR decline from 52.20% to 41.47%. Qwen2.5-32B and Gemma-2-27B-Instruct show the same direction of change; even where CR narrows more modestly (e.g., Gemma 49.43% → 44.30%), pairwise non-transitivity is still more than halved (48.76% → 22.41%). Taken together, the improvements persist when quality is decomposed into orthogonal components rather than measured as a single undifferentiated score.

Mechanistically, the scalar channel benefits from distribution-sensitive scoring, which smooths discretization artifacts and reduces clashes between numeric scores and pairwise preferences, lowering CR. The pairwise channel benefits from likelihood-aware aggregation with calibrated tie handling, which suppresses position bias, lowering NTR. Because these effects arise within each dimension before averaging, the evidence indicates genuine generalization of TrustJudge to multi-dimensional evaluation.

## G GENERALIZATION ACROSS DATASET CATEGORIES

**Setup.** To assess whether our observations generalize across task types, we used 120 prompts from MT-Bench and Arena Hard as the main experiment; for each prompt we independently collected ten model responses so as to obtain a quality-diverse distribution of outputs, yielding a total of 1,200 responses. The 120 prompts were assigned to eight predefined MT-Bench categories as shown in Figure 11. Evaluation was performed with three judges — Qwen2.5-7B-Instruct, Llama-3.1-8B-Instruct, and Gemma-2-9b-it — which each assessed all 1,200 responses using both (i) a single-score assessment on a 5-point scale and (ii) pairwise comparisons between responses. Per category we report the Conflict Ratio under the 5-point and the Non-Transitivity Ratio with $k = 4$. Results are presented both aggregated across categories and broken down by the eight MT-Bench categories to illustrate the variation in inconsistency patterns across task types.

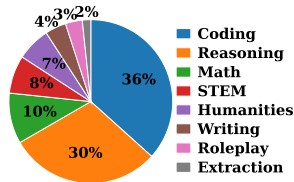

Figure 11: Category distribution across the eight dataset categories.

**Results and analysis.** As shown in Table 6, across eight MT-Bench categories and three judges, the clearest pattern is in pairwise transitivity consistency: non-transitivity ratio collapses from 18.74% under the two-pass baseline to 4.40% with likelihood-aware aggregation method and 5.64% with the PPL-based method (averaged over all 24 category–judge cells). That reduction is uniform—almost every category and every judge shows single-digit NTR after applying our pairwise aggregation, with extremes such as Llama–STEM reaching 0.00%, and large cuts in difficult regimes like Qwen–Math (32.85% → 4.46%). In short, once responses are compared bidirectionally with likelihood-aware tie handling, residual inconsistencies are rare regardless of task type.

Score-Comparison Conflicts show a more nuanced, category-dependent story. Averaged over all cells, CR drops from 23.32% to 20.63% with distribution-sensitive scoring. However, looking category-wise, our method is the best (or tied best) in three of eight groups that emphasize open-ended generation—Coding (Ours 21.78% vs. G-Eval 22.13%), Reasoning (Ours 20.72% vs. G-Eval 21.17%), and Writing (Ours 23.93% vs. G-Eval 24.09%)—while G-Eval leads in STEM, Humanities, Roleplay, and Extraction. Math is the lone case where the raw baseline edges out both methods by a small margin (Baseline 19.41% vs. Ours 19.55%/ G-Eval 20.10%). These contrasts suggest that when responses span a wider stylistic or pragmatic range, TrustJudge that preserves rating entropy tends to reduce score-comparison inconsistency; when the signal is more templated or tightly factual, G-Eval probability summation can be slightly better calibrated.

Table 6: Results for two category-wise inconsistencies. Left block (Score–Comparison Inconsistency): Baseline, G-Eval probability-summation, and TrustJudge's distribution-sensitive scoring on a 5-point scale. Right block (Pairwise Transitivity Inconsistency): two-pass swap-order Baseline, TrustJudge's likelihood-aware aggregation (Option B in 1), and PPL-based method (Option A in 1). Judges are Llama-3.1-8B-Instruct ("Llama"), Qwen2.5-7B-Instruct ("Qwen"), and gemma-2-9b-it ("Gemma").

| Category | Model | Score–Comparison (CR, %) | | | Pairwise Transitivity (NTR$_{k=4}$, %) | | |
|---|---|---|---|---|---|---|---|
| | | **Baseline** | **G-Eval** | **Ours** | **Baseline** | **Likelihood** | **PPL-based** |
| Coding | Llama | 31.19 | 27.74 | **27.59** | 22.07 | **3.72** | 7.80 |
| | Qwen | 26.14 | 25.33 | **23.69** | 19.86 | **4.95** | 6.19 |
| | Gemma | 18.52 | **13.33** | 14.07 | 16.76 | **3.81** | 5.91 |
| Reasoning | Llama | 31.18 | **25.79** | 25.90 | 22.08 | **5.01** | 6.87 |
| | Qwen | 27.53 | 29.35 | **26.48** | 23.93 | **5.56** | 9.69 |
| | Gemma | 10.23 | **8.37** | 9.77 | 14.13 | **2.52** | 5.71 |
| Math | Llama | 24.24 | 25.25 | **24.24** | 23.26 | **4.86** | 5.21 |
| | Qwen | **26.63** | 30.65 | 28.54 | 32.85 | **4.46** | 9.64 |
| | Gemma | 7.35 | **4.41** | 5.88 | 16.84 | **4.29** | 6.48 |
| STEM | Llama | 25.62 | **17.77** | 19.42 | 9.03 | 1.94 | **0.00** |
| | Qwen | 29.35 | 26.75 | **26.23** | 23.07 | **3.68** | 5.70 |
| | Gemma | 9.52 | **4.76** | **4.76** | 9.47 | **1.11** | 3.23 |
| Humanities | Llama | 27.08 | **21.67** | **21.67** | 19.14 | 4.29 | **4.00** |
| | Qwen | 23.88 | 21.49 | **20.30** | 20.38 | 3.86 | **3.41** |
| | Gemma | 12.24 | **2.04** | 6.12 | 7.81 | 2.12 | **1.63** |
| Writing | Llama | 38.71 | **30.97** | **30.97** | 23.10 | **2.07** | 14.83 |
| | Qwen | **20.95** | 30.48 | 30.00 | 26.19 | 10.71 | **5.06** |
| | Gemma | 18.92 | **10.81** | **10.81** | 11.43 | 3.62 | **1.90** |
| Roleplay | Llama | 35.04 | 29.91 | **27.35** | 12.50 | 5.47 | **1.56** |
| | Qwen | 28.49 | **26.16** | 28.49 | 24.69 | **6.76** | 7.70 |
| | Gemma | 16.07 | **5.36** | 14.29 | 10.71 | **4.76** | 6.43 |
| Extraction | Llama | 40.63 | **34.38** | 35.94 | 18.87 | **3.77** | **3.77** |
| | Qwen | 30.12 | **30.12** | 32.53 | 28.85 | **4.23** | 7.69 |
| | Gemma | 0.00 | 0.00 | 0.00 | 12.62 | 8.10 | **5.00** |

Practically, the category study shows the generalization of TrustJudge. The likelihood-aware aggregation and PPL-based method are robust to task type, driving down inconsistencies nearly everywhere. The distribution-sensitive scoring is competitive overall and tends to be strongest where outputs are diverse and rubric-driven (coding, reasoning, writing).

## H  THEORETICAL DERIVATION

### H.1  THEORETICAL ANALYSIS OF DISTRIBUTION-SENSITIVE SCORING

In the LLM-as-a-Judge paradigm, a judge model $M$ assesses a given response $R$. The model's internal assessment can be conceptualized as a conditional probability distribution over a discrete

set of possible scores $\Theta = \{s_1, \ldots, s_k\}$. We denote this probability mass function (PMF) as $p_R(s) \triangleq P_M(S = s|R)$, where $S$ is a random variable representing the score. The uncertainty or ambiguity in this assessment is captured by the conditional entropy:

$$H(S|R) = -\sum_{s \in \Theta} p_R(s) \log p_R(s) \tag{9}$$

Traditional discrete scoring protocols extract a single score by taking the mode of this distribution. We define the discrete scoring function $f_{\text{Discrete}}$ as:

$$f_{\text{Discrete}} : \Delta^{k-1} \to \Theta, \quad f_{\text{Discrete}}(p_R) = \arg\max_{s \in \Theta} p_R(s)$$

where $\Delta^{k-1}$ is the $(k-1)$-simplex representing all possible probability distributions over the $k$ scores. This function maps a probability distribution to a single point estimate. The core issue with this approach is that the $\arg\max$ operator is non-injective; it discards all information about the distribution's shape and uncertainty (entropy), mapping distinct belief states to the same output score. This information loss is a primary source of score-comparison inconsistencies.

In contrast, our proposed distribution-sensitive scoring function, $f_{\text{DS}}$, computes the expected value of the score distribution:

$$f_{\text{DS}} : \Delta^{k-1} \to \mathbb{R}, \quad f_{\text{DS}}(p_R) = \mathbb{E}_{S \sim p_R}[S] = \sum_{s \in \Theta} s \cdot p_R(s)$$

This function maps the entire probability distribution to a continuous scalar value, preserving more information about the underlying assessment. The following theorem formalizes the information preservation property of $f_{\text{DS}}$ compared to the information loss inherent in $f_{\text{Discrete}}$.

**Theorem H.1** (Information Loss of Discrete Scoring and Preservation in Expectation). *Let $p_{R_1}$ and $p_{R_2}$ be two distinct probability distributions over the score set $\Theta$ representing the judge model's assessment of two different responses, $R_1$ and $R_2$ (i.e., $p_{R_1} \neq p_{R_2}$). The discrete scoring function $f_{Discrete}$ can fail to distinguish between these two assessments, whereas the distribution-sensitive scoring function $f_{DS}$ provides a mechanism for their discrimination. Specifically:*

1. *(**Information Loss**): There exist $p_{R_1} \neq p_{R_2}$ with different conditional entropies, $H(S|R_1) \neq H(S|R_2)$, such that their discrete scores are identical: $f_{Discrete}(p_{R_1}) = f_{Discrete}(p_{R_2})$.*

2. *(**Information Preservation**): For the same distributions $p_{R_1}$ and $p_{R_2}$ constructed in (1), their distribution-sensitive scores are distinct: $f_{DS}(p_{R_1}) \neq f_{DS}(p_{R_2})$.*

*Proof.* We will prove the theorem by formal symbolic construction.

Let the score set be $\Theta$. Let us choose two distinct scores $s_m, s_a \in \Theta$ such that $s_m \neq s_a$. Let $s_m$ be the intended mode of our distributions. Further, let us choose two distinct real numbers $\epsilon_1$ and $\epsilon_2$ such that $0 < \epsilon_1, \epsilon_2 < 0.5$ and $\epsilon_1 \neq \epsilon_2$. The condition $\epsilon < 0.5$ ensures that $1 - \epsilon > \epsilon$, which will be necessary to establish $s_m$ as the unique mode. The condition $\epsilon_1 \neq \epsilon_2$ ensures the resulting distributions are distinct.

Consider two responses, $R_1$ and $R_2$, which elicit two different internal belief distributions from the judge model, defined as follows:

1. Let $p_{R_1}$ be a probability mass function (PMF) where the probability mass is concentrated on $s_m$ and $s_a$:

$$p_{R_1}(s) = \begin{cases} 1 - \epsilon_1 & \text{if } s = s_m \\ \epsilon_1 & \text{if } s = s_a \\ 0 & \text{otherwise} \end{cases}$$

2. Let $p_{R_2}$ be a second, distinct PMF, also concentrated on $s_m$ and $s_a$ but with a different balance:

$$p_{R_2}(s) = \begin{cases} 1 - \epsilon_2 & \text{if } s = s_m \\ \epsilon_2 & \text{if } s = s_a \\ 0 & \text{otherwise} \end{cases}$$

Since $\epsilon_1 \neq \epsilon_2$, we have $p_{R_1} \neq p_{R_2}$.

PART 1: PROVING INFORMATION LOSS IN $f_{\text{Discrete}}$

We apply the discrete scoring function $f_{\text{Discrete}}$ to both distributions. By our choice of $\epsilon_1, \epsilon_2 \in (0, 0.5)$, we have $1 - \epsilon_1 > \epsilon_1$ and $1 - \epsilon_2 > \epsilon_2$. Therefore, the mode for both distributions is uniquely $s_m$:

$$f_{\text{Discrete}}(p_{R_1}) = \arg\max_{s \in \Theta} p_{R_1}(s) = s_m$$
$$f_{\text{Discrete}}(p_{R_2}) = \arg\max_{s \in \Theta} p_{R_2}(s) = s_m$$

Thus, we have shown that for two distinct distributions $p_{R_1}$ and $p_{R_2}$, it is possible that $f_{\text{Discrete}}(p_{R_1}) = f_{\text{Discrete}}(p_{R_2})$.

Now, we consider their conditional entropies. The entropy of these distributions is a function of $\epsilon$:

$$H(S|R_1) = -((1 - \epsilon_1) \log(1 - \epsilon_1) + \epsilon_1 \log \epsilon_1)$$
$$H(S|R_2) = -((1 - \epsilon_2) \log(1 - \epsilon_2) + \epsilon_2 \log \epsilon_2)$$

The binary entropy function $H(p) = -p \log p - (1 - p) \log(1 - p)$ is strictly increasing on the interval $(0, 0.5)$. Since we chose $\epsilon_1 \neq \epsilon_2$ within this interval, it follows that $H(S|R_1) \neq H(S|R_2)$. This confirms that $f_{\text{Discrete}}$ maps distributions with different levels of uncertainty to the same output, thereby losing information. This proves the first part of the theorem.

PART 2: PROVING INFORMATION PRESERVATION IN $f_{\text{DS}}$

Next, we apply the distribution-sensitive scoring function $f_{\text{DS}}$ to the same distributions $p_{R_1}$ and $p_{R_2}$:

$$f_{\text{DS}}(p_{R_1}) = \mathbb{E}[S|R_1] = \sum_{s \in \Theta} s \cdot p_{R_1}(s) = s_m(1 - \epsilon_1) + s_a(\epsilon_1)$$
$$f_{\text{DS}}(p_{R_2}) = \mathbb{E}[S|R_2] = \sum_{s \in \Theta} s \cdot p_{R_2}(s) = s_m(1 - \epsilon_2) + s_a(\epsilon_2)$$

To demonstrate that their scores are distinct, let us assume for contradiction that they are equal:

$$f_{\text{DS}}(p_{R_1}) = f_{\text{DS}}(p_{R_2})$$
$$s_m(1 - \epsilon_1) + s_a(\epsilon_1) = s_m(1 - \epsilon_2) + s_a(\epsilon_2)$$
$$s_m - s_m\epsilon_1 + s_a\epsilon_1 = s_m - s_m\epsilon_2 + s_a\epsilon_2$$
$$\epsilon_1(s_a - s_m) = \epsilon_2(s_a - s_m)$$
$$(\epsilon_1 - \epsilon_2)(s_a - s_m) = 0$$

This equality can only hold if $\epsilon_1 - \epsilon_2 = 0$ or $s_a - s_m = 0$. However, by our initial construction, we chose $\epsilon_1 \neq \epsilon_2$ (so $\epsilon_1 - \epsilon_2 \neq 0$) and $s_a \neq s_m$ (so $s_a - s_m \neq 0$). This leads to a contradiction.

Therefore, our assumption must be false, and it must be that $f_{\text{DS}}(p_{R_1}) \neq f_{\text{DS}}(p_{R_2})$. The distribution-sensitive scoring function successfully distinguishes between these two belief states, preserving the discriminative information lost by $f_{\text{Discrete}}$. This proves the second part of the theorem. $\square$

## H.2 THEORETICAL ANALYSIS OF LIKELIHOOD-AWARE AGGREGATION

The PPL-based estimator is designed to resolve ambiguity. From an information-theoretic perspective, ambiguity in a discrete choice problem corresponds to a high-entropy probability distribution over the possible outcomes. The PPL-based method leverages an alternative signal—the generative likelihood of the rationale—to induce a more confident (lower-entropy) posterior belief for decision-making. The following proposition formalizes this concept.

**Proposition H.2** (Uncertainty Reduction via PPL-based Method). *Let $\mathcal{C} = \{1, -1, 0\}$ be the set of outcomes. Let $p(C|\pi)$ be the original outcome distribution from the judge model, and $H(C|\pi)$ its Shannon entropy. In an **ambiguous regime**, this distribution approaches uniformity, causing $H(C|\pi) \rightarrow \log |\mathcal{C}|$.*

*Let a new "confidence" distribution $p_{conf}$ be derived from the rationales $J_k$ for each outcome $k \in \mathcal{C}$:*

$$p_{conf}(k) = \frac{\exp(-\gamma \cdot PPL(J_k))}{\sum_{i \in \mathcal{C}} \exp(-\gamma \cdot PPL(J_i))}$$

*where $\gamma > 0$ is a scaling constant. If there exists at least one outcome $k$ whose rationale has a strictly lower perplexity than another (i.e., $\exists k_1, k_2$ s.t. $PPL(J_{k_1}) < PPL(J_{k_2})$), then there exists a $\gamma$ such that the entropy of the confidence distribution is strictly lower than the maximum possible entropy:*

$$H(p_{conf}) < \log |\mathcal{C}|$$

*This demonstrates that $\hat{C}_{PPL}$ makes a decision based on a more certain signal, reducing the judgment uncertainty present in the original ambiguous distribution.*

*Proof.* The Shannon entropy function, $H(p) = -\sum_i p_i \log p_i$, is a strictly concave function over the probability simplex. Its unique maximum is achieved when the distribution $p$ is uniform, i.e., $p_i = 1/|\mathcal{C}|$ for all $i$. In this case, $H(p) = \log |\mathcal{C}|$.

In an ambiguous regime, the original outcome distribution $p(C|\pi)$ is, by definition, near-uniform. Consequently, its entropy $H(C|\pi)$ is close to its maximum possible value, $\log |\mathcal{C}|$.

Now, consider the confidence distribution $p_{\text{conf}}$. The condition $\exists k_1, k_2$ s.t. $PPL(J_{k_1}) < PPL(J_{k_2})$ implies that the values $\exp(-\gamma \cdot PPL(J_k))$ are not all equal. As a result, after normalization, the distribution $p_{\text{conf}}$ is **not uniform**.

Because the Shannon entropy function's maximum is uniquely attained by the uniform distribution, any non-uniform distribution must have a strictly lower entropy. Therefore,

$$H(p_{\text{conf}}) < \max_p H(p) = \log |\mathcal{C}|$$

Since $H(C|\pi) \approx \log |\mathcal{C}|$, it follows that $H(p_{\text{conf}}) < H(C|\pi)$.

The parameter $\gamma$ controls the "peakedness" of $p_{\text{conf}}$. As $\gamma \to \infty$, $p_{\text{conf}}$ approaches a Kronecker delta function centered at the outcome with the minimum PPL, and its entropy approaches zero. Thus, for any non-trivial difference in PPLs, we can always choose a $\gamma$ to make the decision signal arbitrarily certain.

This proves that the PPL-based method transforms a high-entropy (ambiguous) belief state into a lower-entropy (more certain) one, thereby providing a more discriminative signal for making a final judgment. $\qquad\square$

A desirable property of any comparison function $f(R_x, R_y)$ is symmetry, meaning that swapping the inputs should simply invert the outcome, i.e., $f(R_y, R_x) = -f(R_x, R_y)$. Single-pass estimators often violate this property due to positional bias. The following proposition proves that our bidirectional estimator is inherently stable and symmetric by construction.

**Proposition H.3** (Symmetry and Stability of the Bidirectional Estimator)**.** *Let the single-pass greedy estimator be $\hat{C}_{SP}(R_x, R_y) = \arg\max_k p(k|(R_x, R_y), \mathcal{M})$. Due to positional bias, this estimator is not generally symmetric, meaning there exist pairs $(R_x, R_y)$ for which $\hat{C}_{SP}(R_x, R_y) \neq -\hat{C}_{SP}(R_y, R_x)$.*

*In contrast, the bidirectional estimator $\hat{C}_B$ is **perfectly symmetric** for all inputs:*

$$\hat{C}_B(R_x, R_y) = -\hat{C}_B(R_y, R_x)$$

*This property makes $\hat{C}_B$ a stable estimator with respect to the input ordering.*

*Proof.* Let's define the aggregated score function for $\hat{C}_B(R_x, R_y)$ as $m(k; R_x, R_y) = p(k|(R_x, R_y)) + p(-k|(R_y, R_x))$. The decision is $\arg\max_k m(k; R_x, R_y)$.

Now consider the estimator for the swapped input, $\hat{C}_B(R_y, R_x)$. Its score function is $m(k; R_y, R_x) = p(k|(R_y, R_x)) + p(-k|(R_x, R_y))$.

Let's compare the score for outcome $k = 1$ in the first case with the score for outcome $k = -1$ in the second case:

$$m(1; R_x, R_y) = p(1|(R_x, R_y)) + p(-1|(R_y, R_x))$$
$$m(-1; R_y, R_x) = p(-1|(R_y, R_x)) + p(-(-1)|(R_x, R_y)) = p(-1|(R_y, R_x)) + p(1|(R_x, R_y))$$

We see that $m(1; R_x, R_y) = m(-1; R_y, R_x)$. By the same logic, $m(-1; R_x, R_y) = m(1; R_y, R_x)$ and $m(0; R_x, R_y) = m(0; R_y, R_x)$.

This means that the score assigned to preference "$R_x \succ R_y$" in the first ordering is identical to the score assigned to preference "$R_x \succ R_y$" (which is outcome $-1$) in the second ordering. Therefore, if the maximum score in the first case is for outcome $k^*$, the maximum score in the second case must be for outcome $-k^*$. This proves $\hat{C}_B(R_x, R_y) = -\hat{C}_B(R_y, R_x)$. $\square$

