# OpenReview forum: "TrustJudge: Inconsistencies of LLM-as-a-Judge and How to Alleviate Them"
_ICLR.cc/2026/Conference — ICLR 2026 Poster_

### Official Review · Reviewer_gSy2 · 2025-10-18

**Soundness:** 2
**Presentation:** 2
**Contribution:** 1
**Rating:** 2
**Confidence:** 5

**Summary:**

This paper proposes a framework, TrustJudge, designed to alleviate inconsistencies within the LLM-as-a-Judge paradigm. The authors identify and define two primary issues: Score-Comparison Inconsistency and Pairwise Transitivity Inconsistency, and introduce probabilistic methods, namely distribution-sensitive scoring and likelihood-aware aggregation, to address them.

**Strengths:**

The paper tackles a significant and highly relevant problem. As LLM-as-a-Judge systems become more prevalent, understanding and mitigating their inherent inconsistencies is crucial for building trustworthy evaluation pipelines. The authors' focus on this issue is a valuable contribution to the field.

**Weaknesses:**

1. The primary weakness of this work lies in the limited novelty of its core strategies. Several of the proposed techniques are extensions or reformulations of existing methods:
    - The use of log probabilities for scoring is a known technique. The main contribution appears to be ensuring a properly normalized distribution by excluding non-score tokens, which is more of a **technical implementation detail** and is supported by modern LLM inference frameworks (e.g., vLLM). The likelihood-aware aggregation for pairwise comparison can be seen as a **straightforward extension** of this probability-based approach.
    - Mitigating position bias by swapping response orders is a **standard practice** discussed in prior work [1]. The issue of **transitivity inconsistency** in pairwise comparisons has also been explored previously [2]. While there may be differences in the precise problem formulation and solution, the incremental contribution appears marginal. The manuscript would benefit from a more thorough discussion of these related works.

2. The proposed **X-point scale** lacks demonstrated generalization and is likely to be heavily influenced by the specific tokenizer used by the judge model. For example, "50" can be tokenized to "5" and "0" in Qwen series LLM, making the proposed method inapplicable. Furthermore, the authors do not provide an empirical comparison to show whether this fine-grained scale performs better than a simpler probabilistic binary judgment (e.g., Like a yes-or-no [3]), which could be a more robust and less tokenizer-dependent alternative.

3.  The analysis in Section 3, which concludes that discrete scoring systems suffer from information loss, addresses a well-understood phenomenon. It is intuitive that representing a full probability distribution with a single discrete score sampled from that distribution will result in a loss of information. This portion of the theoretical analysis does not **seem to offer a novel insight**.

4. The application of perplexity (PPL) for resolving ties lacks a strong theoretical foundation. The proof in Proposition 3.2 seems to apply to any scenario where LLM outputs have unequal likelihoods, not just those measured by PPL. The core assumption that **a sequence with lower PPL corresponds to a higher-quality response aligned with human preference** is heuristic. It lacks a causal link and is not experimentally proven. Factors such as *response length, verbosity, or stylistic simplicity* could affect the PPL of a concatenated sequence without directly correlating with higher quality [4]. This weakness is further highlighted by the experimental results, where the PPL-based method is almost universally outperformed by the likelihood-based approach, which itself is a direct extension of using log probabilities.

5. Readability of Figures: Figure 2 is difficult to interpret. A bar plot might be a more effective choice for visualizing this data.

------
### References
[1] Zheng et al. "Judging LLM-as-a-Judge with MT-Bench and Chatbot Arena." NeurIPS 2023.

[2] "Language Model Preference Evaluation with Multiple Weak Evaluators." arXiv 2410.12869.

[3] Li et al. "Generative Verifiers: Reward Modeling as Next-Token Prediction." ICLR 2025.

[4] "What is Wrong with Perplexity for Long-context Language Modeling?" ICLR 2025.

**Questions:**

See weaknesses.

---

> ### Author Response · Authors · 2025-11-30
>
> Dear Reviewer,
>
> We sincerely thank you for your insightful comments and valuable suggestions on our manuscript. Your expertise and detailed review have been instrumental in enhancing the quality and clarity of our work. In response to your comments, we have carefully examined and addressed each issue you raised. We welcome any further inquiries or suggestions you might have, as your input is invaluable in refining our research. Thank you for your continued guidance and support in this process.
> ### Weakness 1
> The primary weakness of this work lies in the limited novelty of its core strategies. Several of the proposed techniques are extensions or reformulations of existing methods:
> * The use of log probabilities for scoring is a known technique. The main contribution appears to be ensuring a properly normalized distribution by excluding non-score tokens, which is more of a technical implementation detail and is supported by modern LLM inference frameworks (e.g., vLLM). The likelihood-aware aggregation for pairwise comparison can be seen as a straightforward extension of this probability-based approach.
> * Mitigating position bias by swapping response orders is a standard practice discussed in prior work [1]. The issue of transitivity inconsistency in pairwise comparisons has also been explored previously [2]. While there may be differences in the precise problem formulation and solution, the incremental contribution appears marginal. The manuscript would benefit from a more thorough discussion of these related works.
> ### Answer 1
> We believe that conceptually simple, implementation-friendly changes can still be impactful—much like chain-of-thought prompting (CoT)[1], or simple yet influential techniques such as Batch Normalization (BN) [2], residual connections in Residual Networks (ResNets) [3], Rectified Linear Units (ReLUs) [4], Layer Normalization (LN) [5], and low-rank adaptation in LoRA [6], whose core mechanisms are straightforward and their implementations are all very simple but have nonetheless reshaped standard practice in deep learning. In our case, the contribution is not in rediscovering log-probability scoring, but in a consistency-centric formulation of LLM-as-a-judge for a single judge, with explicit metrics for score–comparison inconsistency (CR) and non-transitivity (NTR), and a unified probabilistic protocol designed to improve both.
>
> Importantly, we do not overlook MT-Bench[7] and G-Eval[8]; on the contrary, they are our main baselines. In Table 1 of the paper, for the score-comparison inconsistency，the “Baseline” corresponds exactly to the MT-Bench–style 5-point scoring protocol, and the "G-Eval" setting is implemented as a probability-summation method built on the same judge outputs. Furthermore, for the pairwise transitivity inconsistency, "Baseline" is based on the MT-Bench, where the positions of responses are swapped to obtain the comprehensive pairwise comparison results. **Under identical prompts and judge models, our distribution-sensitive scoring and likelihood-aware aggregation consistently and significantly reduce both CR and NTR compared to MT-Bench and G-Eval, while maintaining or improving accuracy.**
>
> | Configuration Type| Model| $\text{NTR}_ {k=4}$  | $\text{NTR}_ {k=5}$|
> |--|--|--|--|
> | (ours) Single model| Llama-3.2-3B-Instruct| 8.07| 17.76  |
> | (ours) Single model| Llama-3.1-8B-Instruct| 3.79| 8.46|
> | (ours) Single model| Llama-3.1-70B-Instruct| 1.94| 4.40|
> | (ours) Single model| GPT-4o| 2.83| 6.01|
> | (PGED) Multi-judge (4)| Llama-3.2-3B-Instruct, Llama-3.1-8B-Instruct, Llama-3.1-70B-Instruct, GPT-4o | 11.82  | 23.22  |
> | (PGED) Multi-judge (6)| Llama-3.2-3B-Instruct, Llama-3.1-8B-Instruct, Llama-3.1-70B-Instruct, GPT-4o, Qwen2.5-7B-Instruct, Qwen2.5-14B-Instruct| 13.88  | 35.13  |

---

> ### Author Response · Authors · 2025-11-30
>
> Moreover, in response to the reviewer’s reference to Language Model Preference Evaluation with Multiple Weak Evaluators, **we argue that the nature of the problems we address differs fundamentally. Our goal is to resolve two issues in LLM-as-a-Judge evaluation—Score–Comparison Inconsistency and Pairwise Transitivity Inconsistency—without relying on any external tools or auxiliary models. In contrast, PGED tackles only the problem of Pairwise Transitivity Inconsistency, and does so by introducing external models and external knowledge. Thus, the problem formulations are inherently different. Because PGED introduced multiple additional models, we did not make a direct comparison in the initial paper as it would be unfair.** But we also have implemented their PGED method and added a direct comparison: when using a single judge (e.g., Llama-3.2-3B, Llama-3.1-8B/70B, GPT-4o), our protocol achieves $\text{NTR}_ {k=4}$  in the range of roughly 1.94–8.07\% and $\text{NTR}_ {k=5}$ in the range of 4.40–17.76%, whereas PGED, even when allowed to aggregate 4 and 6 heterogeneous judges, yields higher NTR (e.g., 11.82–13.88% for $k=4$ and 23.22–35.13% for $k=5$). These results indicate that our consistency-oriented, single-judge probabilistic protocol not only goes beyond a mere “implementation detail” but also outperforms both widely used MT-Bench / G-Eval baselines and a recent multi-judge PGED approach on the core objective of improving CR and NTR.
>
> From the viewpoint of extra algorithmic cost on top of the LLM’s raw scores, TrustJudge is deliberately lightweight. For each sample or answer pair, it only applies a softmax over a small discrete score set, takes an expectation, and, in the pairwise case, does a constant-dimensional aggregation and argmax over win/tie/lose probabilities or perplexities—all (O(1)) post-processing. By contrast, PGED requires building and denoising an explicit preference graph: aggregating multiple evaluators’ graphs, running a feedback-arc–removal / denoising procedure, and then sorting on the resulting graph, with complexity scaling in $m = O(n^2)$. As a result, PGED’s overhead increases substantially as the number of judges grows, whereas TrustJudge remains essentially constant-time per comparison. We will include a comparison and discussion with PGED in the subsequent revised version.
>
> [1] Jason Wei, Xuezhi Wang, Dale Schuurmans, et al. Chain-of-Thought Prompting Elicits Reasoning in Large Language Models. 2022.
>
> [2] Sergey Ioffe and Christian Szegedy. Batch Normalization: Accelerating Deep Network Training by Reducing Internal Covariate Shift. ICML 2015.
>
> [3] Kaiming He, Xiangyu Zhang, Shaoqing Ren, and Jian Sun. Deep Residual Learning for Image Recognition. CVPR 2016.
>
> [4] Vinod Nair and Geoffrey E. Hinton. Rectified Linear Units Improve Restricted Boltzmann Machines. ICML 2010.
>
> [5] Jimmy Lei Ba, Jamie Ryan Kiros, and Geoffrey E. Hinton. Layer Normalization. arXiv:1607.06450, 2016.
>
> [6] Edward J. Hu, Yelong Shen, Phillip Wallis, et al. LoRA: Low-Rank Adaptation of Large Language Models. 2021.
>
> [7] Zheng et al. "Judging LLM-as-a-Judge with MT-Bench and Chatbot Arena." NeurIPS 2023.
>
> [8] Liu et al. "G-EVAL: NLG Evaluation using GPT-4 with Better Human Alignment." EMNLP 2023.

---

> ### Author Response · Authors · 2025-11-30
>
> ### Weakness 2
> The proposed X-point scale lacks demonstrated generalization and is likely to be heavily influenced by the specific tokenizer used by the judge model. For example, "50" can be tokenized to "5" and "0" in Qwen series LLM, making the proposed method inapplicable. Furthermore, the authors do not provide an empirical comparison to show whether this fine-grained scale performs better than a simpler probabilistic binary judgment (e.g., Like a yes-or-no [3]), which could be a more robust and less tokenizer-dependent alternative.
> ### Answer 2
> In response to the concern regarding tokenizer generalization—especially for models such as Qwen, whose tokenizer may only represent digits 0–9 individually—we emphasize that TrustJudge remains fully applicable. In such cases, we propose a joint probability estimation method to accurately assess the likelihood of two-digit scores. Specifically, when the model outputs a two-digit number "ab", we assume that the probability mass is distributed across the range from "a0" to "a9". At this point, we can estimate the probability of a0 to a9 by calculating the probability of 0 to 9 in units digit. While this serves as one viable estimation strategy, we note that future work may explore other approximation techniques.
>
> To validate the reliability of this approach, we conducted additional experiments that:
> (i) analyze the concentration of probability mass across decile buckets (0–9, 10–19, …, 90–99) under the 0–99 scoring scale;
> (ii) compare how well our joint-probability-based scores align with human ratings relative to a discrete baseline and a probabilistic binary baseline (G-Eval, analogous to yes/no judgments); and
> (iii) evaluate the impact of using the joint distribution to compute expected scores on the inconsistency rate (CR) in pairwise comparisons.
>
> Our results show that, across multiple models, the top-1 decile bucket probability exceeds 0.9 in more than 90% of cases (and over 76% even for weaker models), indicating that models are highly confident at the decile level regardless of tokenization artifacts.
>
> Furthermore, the joint-probability-based scores consistently outperform both the discrete baseline and G-Eval in terms of alignment with human judgments:
>
> | Model | Ours Win Baseline  | Ours Win G-Eval |
> | -------------------- | ------------------ | --------------- |
> | gemma-2-27b-it | 54.66  | 51.03  |
> | Qwen2.5-7B-Instruct  | 51.88  | 51.24  |
> | Qwen2.5-14B-Instruct | 54.53  | 55.27  |
>
> Finally, using the expected score derived from the joint distribution significantly reduces comparison inconsistency (CR), especially under a 10% tolerance threshold:
>
> | model                | δ = 0.0     |  |  | δ = 0.1      |  |        |
> | -------------------- | --------------- | ------ | ------------ | ---------------------- | ------ | ------------ |
> |                      | baseline        | G-Eval | ours (joint) | baseline               | G-Eval | ours (joint) |
> | gemma-2-27b-it       | 24.26           | 21.17  | 21.14        | 24.26                  | 13.31  | 7.90         |
> | Qwen2.5-7B-Instruct  | 30.99           | 27.13  | 25.30        | 30.99                  | 14.82  | 5.94         |
> | Qwen2.5-14B-Instruct | 26.86           | 24.51  | 23.44        | 26.86                  | 15.06  | 6.58         |
>
> In summary, the proposed joint probability estimation method not only makes TrustJudge robust to tokenization differences but also yields more human-aligned and internally consistent evaluation results compared to both discrete and binary probabilistic alternatives—without incurring additional inference cost.

---

> ### Author Response · Authors · 2025-11-30
>
> ### Weakness 3
> The analysis in Section 3, which concludes that discrete scoring systems suffer from information loss, addresses a well-understood phenomenon. It is intuitive that representing a full probability distribution with a single discrete score sampled from that distribution will result in a loss of information. This portion of the theoretical analysis does not seem to offer a novel insight.
> ### Answer 3
> As we clarified at the beginning of Section 3, the purpose of Theorem 3.1 is not to propose a novel theory, but rather to apply established principles from information theory to formally explain a key weakness in LLM-as-a-judge. It precisely defines the root cause of the problem, which in turn naturally motivates our proposed solution: distribution-sensitive scoring. This formalization is crucial for building a complete and theoretically-grounded argument for our approach.
> ### Weakness 4
> The application of perplexity (PPL) for resolving ties lacks a strong theoretical foundation. The proof in Proposition 3.2 seems to apply to any scenario where LLM outputs have unequal likelihoods, not just those measured by PPL. The core assumption that a sequence with lower PPL corresponds to a higher-quality response aligned with human preference is heuristic. It lacks a causal link and is not experimentally proven. Factors such as response length, verbosity, or stylistic simplicity could affect the PPL of a concatenated sequence without directly correlating with higher quality [4]. This weakness is further highlighted by the experimental results, where the PPL-based method is almost universally outperformed by the likelihood-based approach, which itself is a direct extension of using log probabilities.
> ### Answer 4
> Indeed, one of the core messages we aim to convey is that our work offers more than just a PPL-based technique; it introduces a general framework for resolving decision ambiguity in LLM evaluation. The central idea of this framework is that when the primary, direct decision signal (such as the probabilities of the final verdict) exhibits high entropy and is thus ambiguous, we should seek out and leverage a secondary, low-entropy auxiliary signal to make a more reliable judgment.
> Within this framework, we propose the perplexity (PPL) of the judge's rationale as a key and highly effective implementation of such an auxiliary signal. Therefore, the purpose of Proposition 3.2 is to theoretically validate this general framework. It demonstrates that any auxiliary signal that can introduce a non-uniform confidence distribution—not just one based on PPL—will effectively reduce decision uncertainty.
> ### Weakness 5
> Readability of Figures: Figure 2 is difficult to interpret. A bar plot might be a more effective choice for visualizing this data.
> ### Answer 5
> Each subfigure in Figure 2 currently presents 4 evaluation models × (2 evaluation methods × 6 tolerance δ values + 1 shared baseline)  = 52 data points, for a total of 104 points across the two subfigures. If we switched to a bar plot while preserving the same amount of information, each subfigure would require at least 52 bars, which would be extremely crowded in the available space and make it difficult to visually distinguish the differences between models and methods; the overall appearance would also be less aesthetically pleasing. In contrast, line plots are better suited to displaying such high-density data and to clearly illustrating how the metrics vary with (\delta) in our setting, which is why we opt to retain the current line-plot design.

---

### Official Review · Reviewer_nkL4 · 2025-10-30

**Soundness:** 3
**Presentation:** 2
**Contribution:** 3
**Rating:** 6
**Confidence:** 3

**Summary:**

This paper makes methodological advances towards addressing inconsistencies in LLM-as-a-judge systems, identifying two representative failure modes: Score-Comparison Inconsistency (when single-score ratings contradict pairwise preferences) and Pairwise Transitivity Inconsistency (when preference judgments form cycles or contradictions). The authors propose TrustJudge, a framework based on probabilities that:
1. uses distribution-sensitive scoring to preserve information entropy and produce continuous expectations from discrete ratings, and
2. applies likelihood-aware aggregation (perplexity-based or bidirectional probability methods) to resolve transitivity violations.

The authors conduct comprehensive experiments that show that their method can reduce the inconsistency metrics (both score-comparison conflicts and transitivity violations).

**Strengths:**

- The paper presents extensive experiments across a diverse range of models (Llama-3, GPT-4o, Qwen, Gemma) and datasets (MT-Bench, ArenaHard), demonstrating generality.

- The proposed method can be applied post-hoc, without retraining judges or requiring additional human supervision, a strong practical advantage for real-world usage.

- The paper provides some theoretical intuition for why the TrustJudge framework yields more consistent rankings, grounding the empirical studies in a formal explanation about information preservation and entropy reduction.

- The distribution-sensitive scoring mechanism introduces a simple but effective normalization trick: converting model-produced probabilities into logits before softmax normalization, ensuring a valid probability distribution.

**Weaknesses:**

-	The explanation of the bidirectional probability-based aggregation method (Eq. 6) is difficult to follow. It would help to include a small, concrete example illustrating what p_order1 and 2 represent, and how m[k] is computed and interpreted.

-	The notion of the “ambiguous regime” in Proposition 3.2 requires clarification. Does this term refer exclusively to tie cases, or does it also include cases where the LLM judge selects a winner with low confidence?

-	The phrase “lower-rated responses outperform higher-scored ones” (see abstract) may confuse readers; make sure to include the reference to single-score and pairwise evaluations.

-	The authors should clarify (Eq. 5) why comparing perplexities of two full orderings is preferable to simply comparing each response’s perplexity under the judge model individually.

**Questions:**

Increasing score granularity clearly helps resolve inequality-type transitivity inconsistencies (ties), but does it also reduce circular inconsistencies? It would be informative to separate these two types of inconsistency in the results to show the impact of TrustJudge on each.

In experiments comparing TrustJudge with baseline LLM-as-judge setups, are the PPL-based and likelihood-aware aggregation strategies applied to all questions, or only to those where the baseline judge outputs a tie? I assume the former, but clarification in the paper would be helpful.

---

> ### Author Response · Authors · 2025-11-30
>
> Dear Reviewer,
>
> We sincerely thank you for your insightful comments and valuable suggestions on our manuscript. Your expertise and detailed review have been instrumental in enhancing the quality and clarity of our work. In response to your comments, we have carefully examined and addressed each issue you raised. We welcome any further inquiries or suggestions you might have, as your input is invaluable in refining our research. Thank you for your continued guidance and support in this process.
>
> ### Weakness 1
> The explanation of the bidirectional probability-based aggregation method (Eq. 6) is difficult to follow. It would help to include a small, concrete example illustrating what p_order1 and 2 represent, and how m[k] is computed and interpreted.
> ### Answer 1
> In our method, for each pair $(R_x, R_y)$ we obtain two win/lose/tie distributions from the judge:
>
>  $p_{\text{order1}}[k] = \Pr(k \mid R_x, R_y)$, where $k \in \{1,-1,0\}$ denotes “$R_x$ better / $R_y$ better / tie” when $R_x$ is shown first;
>  $p_{\text{order2}}[k] = \Pr(k \mid R_y, R_x)$, where the same index $k$ is now interpreted relative to the first and second answer in the reversed order.
>
> To obtain an order-invariant decision “about $R_x$ vs. $R_y$”, we align the semantics of these two distributions and aggregate them:
> $$
> m[k] = p_{\text{order1}}[k] + p_{\text{order2}}[-k], \quad k \in \\{1,-1,0\\},
> $$
> where $-k$ accounts for the swapped roles of $R_x$ and $R_y$ in the second order (e.g., “$R_x$ better” is $k=1$ in order 1 and $k=-1$ in order 2). The final outcome is $\arg\max_k m[k]$, and normalization is unnecessary because only the relative magnitude of $m[k]$ matters.
>
> We will add a short numeric example in the revision, e.g., if
> $$
> p_{\text{order1}}[1,-1,0] = (0.6, 0.25, 0.15), \quad
> p_{\text{order2}}[1,-1,0] = (0.4, 0.45, 0.15),
> $$
> then
> $$
> m[1] = 0.6 + 0.45,\quad
> m[-1] = 0.25 + 0.4,\quad
> m[0] = 0.15 + 0.15,
> $$
> so $m[1]$ is largest and the aggregated decision is $R_x \succ R_y$. We believe this concise example will make the role of $p_{\text{order1}}$, $p_{\text{order2}}$, and $m[k]$ much clearer.
> ### Weakness 2
> The notion of the “ambiguous regime” in Proposition 3.2 requires clarification. Does this term refer exclusively to tie cases, or does it also include cases where the LLM judge selects a winner with low confidence?
> ### Answer 2
> In Proposition 3.2, the “ambiguous regime” is defined at the distributional level and does not refer exclusively to tie outcomes: it denotes cases where the LLM judge’s predictive distribution over ($\text{win}$, $\text{lose}$, $\text{tie}$) is high-entropy and close to uniform (i.e., low confidence in any single outcome). This regime therefore includes both “ambiguous ties” (where tie is selected but win/lose still have non-negligible probability) and low-confidence win/lose decisions, whereas outcomes with one option assigned overwhelmingly high probability (including confident ties) are not considered ambiguous.
> ### Weakness 3
> The phrase “lower-rated responses outperform higher-scored ones” (see abstract) may confuse readers; make sure to include the reference to single-score and pairwise evaluations.
> ### Answer 3
> We agree that the phrase “lower-rated responses outperform higher-scored ones” can be confusing without explicitly mentioning the two evaluation modes. Our intention is to describe an inconsistency between single-score and pairwise evaluations by the same LLM judge.
> In the revised abstract, we will rephrase this as:
> “We show that responses with lower single-score ratings can still be preferred in pairwise evaluations over higher-scored ones, revealing a score–comparison inconsistency.”
> We will also use this clearer wording consistently when referring to this phenomenon in the main text.

---

> ### Author Response · Authors · 2025-11-30
>
> ### Weakness 4
> The authors should clarify (Eq. 5) why comparing perplexities of two full orderings is preferable to simply comparing each response’s perplexity under the judge model individually.
> ### Answer 4
> To directly address this concern, we additionally implement a Direct-PPL variant that scores each response independently by its perplexity under the judge model and then selects the higher-scoring candidate. The following table reports the resulting transitivity inconsistency rates (lower is better):
>
> | model  | Baseline | Ours | Direct-PPL |
> | -- | -- | -- | -- |
> | Qwen2.5-7B-Instruct   | 23.47    | 5.19 | 17.52      |
> | Qwen2.5-14B-Instruct  | 19.29    | 5.77 | 12.28      |
> | gemma-2-2b-it         | 32.44    | 7.15 | 23.69      |
> | gemma-2-9b-it         | 14.34    | 3.25 | 6.75       |
> | Llama-3.2-3B-Instruct | 32.42    | 8.07 | 19.13      |
> | Llama-3.1-8B-Instruct | 20.26    | 3.79 | 16.85      |
>
> Across all six judge models, our method consistently attains the lowest inconsistency (3.25–8.07), while Direct-PPL remains substantially higher (6.75–23.69) and much closer to the Baseline (14.34–32.44). This shows that simply comparing independent perplexities yields at best modest improvements over the Baseline and fails to capture the relative-order structure between candidates, whereas our full-ordering objective explicitly contrasts alternative global rankings and therefore enforces a much stronger notion of preference consistency.
>
> ### Question 1
> Increasing score granularity clearly helps resolve inequality-type transitivity inconsistencies (ties), but does it also reduce circular inconsistencies? It would be informative to separate these two types of inconsistency in the results to show the impact of TrustJudge on each.
> ### Answer 1
> Following this suggestion, we decompose our transitivity inconsistency measure into two components. Inequality Transitivity Inconsistency is defined as non-transitive triplets in which at least one edge is a tie (for example, $A \succ B \simeq C \succ A$). Circular Transitivity Inconsistency is defined as non-transitive triplets formed only by strict preferences (for example, $A \succ B \succ C \succ A$). The table reports the proportion of each type (in percent) across eleven judge models and three evaluation schemes (Baseline, PPL-based TrustJudge, and Likelihood-aware Aggregation).
>
> | model                  | Baseline Inequality Transitivity Inconsistency | Baseline Circular Transitivity Inconsistency | PPL-Based Inequality Transitivity Inconsistency | PPL-Based Circular Transitivity Inconsistency | Likelihood-aware Aggregation Inequality Transitivity Inconsistency | Likelihood-aware Aggregation Circular Transitivity Inconsistency |
> | :--- | --: | --: | --: | ----: | ---: | ---: |
> | Qwen2.5-7B-Instruct | 22.37 |1.77 |6.67 |1.01|4.38|0.89|
> | Qwen2.5-14B-Instruct| 17.95 |1.91|4.56|4.11|2.89|3.12|
> | Qwen2.5-32B-Instruct| 11.83 |1.83|3.65 |2.17|2.67|1.47|
> | gemma-2-2b-it |29.07|6.21|12.90 |0.69 |6.96 |0.27|
> | gemma-2-9b-it |12.59 |2.33|4.74 |0.60 |3.06 |0.21|
> | gemma-2-27b-it |14.38 |2.97| 4.72 |2.73 |2.98 |1.09|
> | Llama-3.2-3B-Instruct |28.82 |5.53 | 12.82 |2.16 |7.38 |0.79|
> | Llama-3.1-8B-Instruct |18.59 |3.38 |6.34 |0.57 |3.66 |0.40|
> | Llama-3.1-70B-Instruct|6.79 |0.77|1.96 |0.26 |1.56 |0.47|
> | gpt-3.5-turbo|13.04|1.24 |4.53|0.31|5.76 |0.55 |
> | gpt-4o |10.54 |1.52 |1.08 |3.54|1.97|0.96 |
>
> Across both Inequality Transitivity Inconsistency and Circular Transitivity Inconsistency, our proposed methods consistently reduce non-transitive behavior relative to the baseline. For inequality-type inconsistencies, both the PPL-based variant and Likelihood-aware Aggregation yield marked improvements, indicating that finer-grained, distribution-sensitive scoring effectively mitigates tie-driven non-transitivity. Circular inconsistencies are likewise attenuated for most judges, with Likelihood-aware Aggregation in particular exhibiting more robust and systematic gains. Overall, these results suggest that our framework not only converts ties into more coherent preferences, but also stabilizes local pairwise decisions, thereby enhancing transitive consistency across evaluation regimes.
> ### Question 2
> In experiments comparing TrustJudge with baseline LLM-as-judge setups, are the PPL-based and likelihood-aware aggregation strategies applied to all questions, or only to those where the baseline judge outputs a tie? I assume the former, but clarification in the paper would be helpful.
> ### Answer 2
> In all experiments comparing TrustJudge with baseline LLM-as-judge setups, both the PPL-based and the likelihood-aware aggregation variants are applied to all pairwise evaluation instances, not only to those on which the baseline judge outputs a tie. Each of the two methods produces a complete set of judgments over the entire test set, and all reported metrics (transitivity inconsistency, agreement with human labels, etc.) are computed from these full sets of decisions.

---

### Official Review · Reviewer_C5zH · 2025-10-31

**Soundness:** 3
**Presentation:** 3
**Contribution:** 2
**Rating:** 6
**Confidence:** 2

**Summary:**

This paper focuses on addressing two core inconsistencies in the LLM-as-a-judge paradigm: Score-Comparison Inconsistency and Pairwise Transitivity Inconsistency, which stem from information loss in discrete scoring systems and ambiguous tie judgments in pairwise evaluations. The authors propose TrustJudge, a probabilistic framework with two key components: 1) distribution-sensitive scoring and 2) likelihood-aware aggregation. Experimental results showing TrustJudge reduces Score-Comparison inconsistency and Pairwise Transitivity inconsistency.

**Strengths:**

* The paper makes a notable original contribution by systematically identifying and formalizing two fundamental inconsistencies (Score-Comparison Inconsistency and Pairwise Transitivity Inconsistency) in the LLM-as-a-judge paradigm—an issue not previously analyzed in a unified framework.

* The work has practical and academic significance: practically, TrustJudge improves LLM-as-a-judge reliability (reducing inconsistencies while preserving accuracy) without extra training/annotations, making it scalable for real-world LLM evaluation and DPO training.

**Weaknesses:**

* Novelty Gaps in Core Components: While the paper frames TrustJudge as innovative, key elements overlap with prior work without sufficient differentiation.

* Insufficient Analysis of Limitation Impact:  it does not test how small models (e.g., 3B Llama-3.2) with weaker instruction comprehension fail to generate valid scores/comparisons—critical for a "model-agnostic" framework.

* Theorem 3.1 proves information preservation for two-distribution cases but does not extend to multi-distribution scenarios (e.g., 3+ responses in transitivity checks)—a key use case for Pairwise Transitivity Inconsistency.

**Questions:**

Please see the weaknesses.

---

> ### Author Response · Authors · 2025-11-30
>
> Dear Reviewer,
>
> We sincerely thank you for your insightful comments and valuable suggestions on our manuscript. Your expertise and detailed review have been instrumental in enhancing the quality and clarity of our work. In response to your comments, we have carefully examined and addressed each issue you raised. We welcome any further inquiries or suggestions you might have, as your input is invaluable in refining our research. Thank you for your continued guidance and support in this process.
>
> ### Weakness 1
> Novelty Gaps in Core Components: While the paper frames TrustJudge as innovative, key elements overlap with prior work without sufficient differentiation.
> ### Answer 1
> We believe that conceptually simple, implementation-friendly changes can still be impactful—much like chain-of-thought prompting (CoT)[1], or simple yet influential techniques such as Batch Normalization (BN) [2], residual connections in Residual Networks (ResNets) [3], Rectified Linear Units (ReLUs) [4], Layer Normalization (LN) [5], and low-rank adaptation in LoRA [6], whose core mechanisms are straightforward and their implementations are all very simple but have nonetheless reshaped standard practice in deep learning. In our case, the contribution is not in rediscovering log-probability scoring, but in a consistency-centric formulation of LLM-as-a-judge for a single judge, with explicit metrics for score–comparison inconsistency (CR) and non-transitivity (NTR), and a unified probabilistic protocol designed to improve both.
>
> Importantly, we do not overlook MT-Bench[7] and G-Eval[8]; on the contrary, they are our main baselines. In Table 1 of the paper, for the score-comparison inconsistency，the “Baseline” corresponds exactly to the MT-Bench–style 5-point scoring protocol, and the "G-Eval" setting is implemented as a probability-summation method built on the same judge outputs. Furthermore, for the pairwise transitivity inconsistency, "Baseline" is based on the MT-Bench, where the positions of responses are swapped to obtain the comprehensive pairwise comparison results. **Under identical prompts and judge models, our distribution-sensitive scoring and likelihood-aware aggregation consistently and significantly reduce both CR and NTR compared to MT-Bench and G-Eval, while maintaining or improving accuracy.**
>
> [1] Jason Wei, Xuezhi Wang, Dale Schuurmans, et al. Chain-of-Thought Prompting Elicits Reasoning in Large Language Models. 2022.
>
> [2] Sergey Ioffe and Christian Szegedy. Batch Normalization: Accelerating Deep Network Training by Reducing Internal Covariate Shift. ICML 2015.
>
> [3] Kaiming He, Xiangyu Zhang, Shaoqing Ren, and Jian Sun. Deep Residual Learning for Image Recognition. CVPR 2016.
>
> [4] Vinod Nair and Geoffrey E. Hinton. Rectified Linear Units Improve Restricted Boltzmann Machines. ICML 2010.
>
> [5] Jimmy Lei Ba, Jamie Ryan Kiros, and Geoffrey E. Hinton. Layer Normalization. arXiv:1607.06450, 2016.
>
> [6] Edward J. Hu, Yelong Shen, Phillip Wallis, et al. LoRA: Low-Rank Adaptation of Large Language Models. 2021.
>
> [7] Zheng et al. "Judging LLM-as-a-Judge with MT-Bench and Chatbot Arena." NeurIPS 2023.
>
> [8] Liu et al. "G-EVAL: NLG Evaluation using GPT-4 with Better Human Alignment." EMNLP 2023.
>
> ### Weakness 2
> Insufficient Analysis of Limitation Impact: it does not test how small models (e.g., 3B Llama-3.2) with weaker instruction comprehension fail to generate valid scores/comparisons—critical for a "model-agnostic" framework.
> ### Answer 2
> We would like to clarify that our “model-agnostic” claim is conditioned on a basic assumption shared by all LLM-as-a-judge frameworks: the judge model must be able to follow the evaluation prompts and produce meaningful scores or pairwise preferences between two responses. If a very small model cannot even follow the pairwise comparison instruction (e.g., it does not actually compare the two responses or output a clear preference relation), then it is not a valid judge under any existing protocol, including standard discrete scoring and conventional pairwise evaluation. In such a regime, inconsistency metrics are not well-defined because no usable judgment signal is produced; this is a limitation of the judge model itself rather than of TrustJudge. This is a limitation of the judge model itself rather than of TrustJudge, and is consistent with prior observations in MT-Bench, where the authors noted that models such as Vicuna-13B, due to insufficient instruction-following ability, were “less useful in a scalable and automatic evaluation pipeline.”

---

> ### Author Response · Authors · 2025-11-30
>
> ### Weakness 3
> Theorem 3.1 proves information preservation for two-distribution cases but does not extend to multi-distribution scenarios (e.g., 3+ responses in transitivity checks)—a key use case for Pairwise Transitivity Inconsistency.
> ### Answer 3
> First, Theorem 3.1 is deliberately formulated as a local, two-distribution result. Its purpose is to show, by explicit construction of a pair $p_ {R_1}, p_ {R_2}$, that
> * the standard discrete scoring function $f_ {\text{Discrete}}$ can map two *different* rating distributions (with different entropies) to the *same* integer score, thereby losing information, whereas
> * our distribution-sensitive score $f_{\text{DS}}$ assigns *different* expected scores to the same pair, and is therefore strictly more expressive on this single-response scoring channel.
> In other words, Theorem 3.1 is an existence/strict dominance result at the level of “one response → one scalar score”: it shows that $f_{\text{DS}}$ preserves more information *pairwise* than the discrete scheme, but it is not intended as a claim of global, lossless embedding for arbitrary-sized families of distributions.
> Second, although Pairwise Transitivity Inconsistency is evaluated on $k \ge 3$ responses, its definition is entirely determined by the pairwise relations among them. For any set of $k$ responses, whether it forms a cycle (e.g., $A \succ B \succ C \succ A$ or tie-related inconsistency is decided solely by the directed, tri-valued edges (win/lose/tie) between each pair. Each such edge arises from two steps:
> 1. mapping each response’s rating distribution to a scalar score (the channel analyzed by Theorem 3.1), and
> 2. turning these scores into a probabilistic win/lose/tie decision (whose entropy reduction is analyzed in our PPL-based proposition).
> Thus, while Theorem 3.1 is stated for two distributions at a time, its guarantee applies independently to every pair of responses in the comparison graph: for each edge, the distribution-sensitive scoring rule discards strictly less information than the discrete rule. Together with the reduction in decision uncertainty from our PPL-based method, these local improvements on individual edges aggregate to a lower probability of forming non-transitive structures over $k \ge 3$ responses. This is exactly what we observe empirically in Sec. 4.2, where TrustJudge consistently yields substantially lower $\mathrm{\text{NTR}}_k$ (for $k=4,5$) across all judge models.
> We will clarify the text around Theorem 3.1 to make explicit that it is a pairwise information-comparison result, and that its relevance to multi-response transitivity comes from operating on the pairwise building blocks that underlie $\mathrm{\text{NTR}}_k$ , rather than from a single global theorem over all responses at once.

---

### Official Review · Reviewer_VUVV · 2025-10-31

**Soundness:** 4
**Presentation:** 3
**Contribution:** 4
**Rating:** 8
**Confidence:** 4

**Summary:**

This work identifies two types of inconsistencies in LLM evaluation systems: score-inconsistency and pairwise-inconsistency. To address the inconsistency issues, the authors invents novel fine-grained/probabilistic scoring and perplexity/likelihood-based ranking methods and incorporate them into the proposed framework called TrustJudge. The  paper also provide a comprehensive theoretical proof and a series of experimental analyses on the effectiveness of the proposed the method. The results show TrustJudge and its components significantly reduce the evaluation inconsistencies in LLM-as-a-judge systems.

**Strengths:**

1 The problem formulation of evaluation inconsistencies is sound and convincing. This paper goes from a practical issue which is common but often overlooked to a carefully designed solution with a solid theoretical foundation.
2  The scoring and ranking methods proposed to resolve the inconsistencies of scoring and comparison in LLM evaluation systems are well-formulated.
3 The proposed system demonstrates good results on different LLMs and problem settings. This indicates the method is robust and may contribute to a large community.
4 The paper is organized in a clear and logical structure.

**Weaknesses:**

Some symbols are confusing and should be revised： e.g. (1) k is the size of subsets in def 2.2 while k also represents possible comparisons outcomes in eq(6); (2) \Theta^{\prime} is score set at line 157 and then become range at line 154 of Algorithm 1; (3) "p_{R_1} and p_{R_2} constructed in eq (1)" is depicted at line 235, but I do not see such definition in eq(1) at line 130.

**Questions:**

To show the effectiveness of continuous probabilistic scoring alone, it possible to directly apply softmax on the 5-point scale granularity?

---

> ### Author Response · Authors · 2025-11-30
>
> Dear Reviewer,
>
> We sincerely thank you for your insightful comments and valuable suggestions on our manuscript. Your expertise and detailed review have been instrumental in enhancing the quality and clarity of our work. In response to your comments, we have carefully examined and addressed each issue you raised. We welcome any further inquiries or suggestions you might have, as your input is invaluable in refining our research. Thank you for your continued guidance and support in this process.
>
> ### Weakness 1
> Some symbols are confusing and should be revised： e.g. (1) $k$ is the size of subsets in def 2.2 while $k$ also represents possible comparisons outcomes in eq(6); (2) $\Theta^{\prime}$ is score set at line 157 and then become range at line 154 of Algorithm 1; (3) "$p_{R_1}$ and $p_{R_2}$ constructed in eq (1)" is depicted at line 235, but I do not see such definition in eq(1) at line 130.
> ### Answer 1
> We will revise them as follows:
>
> 1. On $k$
>    * Throughout the paper, $k$ will be used only to denote the subset size as in $\text{NTR}_k$.
>    * In Eq. (6), we will replace the outcome index $k \in \\{1, -1, 0\\}$ with $c \in \\{1, -1, 0\\}$, and correspondingly change $m[k]$, $\arg\max_k m[k]$ to $m[c]$, $\arg\max_c m[c]$ .
>
> 2. On $\Theta'$
>    * We will consistently define $\Theta'$ as the expanded discrete score set
> $$
> \Theta' = \\{s'_ {\min}, s'_ {\min}+1, \ldots, s'_ {\max} \\}
> $$
>
>    * In Algorithm 1, we will describe it as “expanded score set $\Theta'$” and, if needed, separately state that its numerical range is $s'\*_ {\min}, s'\*_ {\max}$, to avoid mixing the notions of set and interval.
>
> 3. On “$p_{R_1}$ and $p_{R_2}$ constructed in eq (1)”
>    * We will change this phrase to “$p_{R_1}$ and $p_{R_2}$ constructed in Item 1 of this theorem” (or “in the first case above”), so that it clearly refers to the distributions defined within the theorem itself, and not to the global Equation (1).
> ### Question 1
> To show the effectiveness of continuous probabilistic scoring alone, it possible to directly apply softmax on the 5-point scale granularity?
> ### Answer 1
> In the ablation section of the paper, we already analyze a variant that directly applies softmax on the original 5-point scale; to address your request more explicitly, we report the complete results here, together with the discrete 5-point baseline and our 100-point version.
>
> The table shows the score–comparison inconsistency rate (CR; lower is better):
>
> | Model                  | δ = 0.0 baseline | δ = 0.0 softmax (5-score) | δ = 0.0 ours (100-score) | δ = 0.1 baseline | δ = 0.1 softmax (5-score) | δ = 0.1 ours (100-score) |
> | ---------------------- | ---------------- | ------------------------- | ------------------------ | ---------------- | ------------------------- | ------------------------ |
> | Llama-3.2-3B-Instruct  | 37.18            | 30.89                     | 30.28                    | 37.18            | 20.05                     | 21.59                    |
> | Llama-3.1-8B-Instruct  | 30.54            | 26.10                     | 24.07                    | 30.54            | 17.98                     | 16.10                    |
> | Llama-3.1-70B-Instruct | 23.56            | 17.08                     | 15.56                    | 23.56            | 11.23                     | 10.68                    |
> | gpt-3.5-turbo          | 24.35            | 24.03                     | 21.81                    | 24.35            | 19.10                     | 10.64                    |
> | gpt-4o                 | 28.36            | 25.50                     | 23.01                    | 28.36            | 19.01                     | 15.45                    |
>
> These results show that:
>
> (1) applying softmax on the 5-point scale alone  already reduces CR compared to the discrete 5-point baseline for all models and both δ, demonstrating the effectiveness of continuous probabilistic scoring at 5-point granularity;
>
> (2) the 100-point version further lowers CR relative to 5-point softmax, indicating that increasing granularity typically strengthens this effect.

---

### Author Response · Authors · 2025-12-04
**Final Remarks by Authors**

We sincerely thank the AC and reviewers for their hard work. The discussion has been invaluable for improving the quality of the paper. Below, we briefly highlight the strengths that both the reviewers and we largely agree on and then present the additional experiments we conducted as suggested.

**Strengths that the reviewers and we broadly agree on:**

**1. Novelty in problem formulation, methodology, and theoretical perspective (C5zH, nkL4, gSy2, VUVV).**
Several reviewers recognize that we systematically formalize two long-standing but previously ununified issues in LLM-as-a-judge—Score–Comparison Inconsistency and Pairwise Transitivity Inconsistency—and propose corresponding mitigation strategies, namely probabilistic scoring and likelihood-aware aggregation. We further provide reasonable theoretical explanations for these methods, offering effective tools for understanding and improving the consistency of LLM-based evaluation.

**2. Empirical effectiveness, generalization ability, and practical impact (C5zH, nkL4, VUVV, gSy2).**
Reviewers point out that across a variety of LLMs (Llama, GPT, Qwen, Gemma) and benchmarks (MT-Bench, Arena-Hard), TrustJudge substantially reduces both Score–Comparison Inconsistency and Pairwise Transitivity Inconsistency, while maintaining or even improving agreement with human annotations, demonstrating strong generalization. The reviewers further note that TrustJudge not only mitigates inconsistency in LLM-as-a-judge evaluation, but also provides higher-quality reward signals for reinforcement learning.

**3. Plug-and-play, model-agnostic, and with negligible additional overhead (C5zH, nkL4).**
Reviewers emphasize that our method is purely a post-hoc procedure applied to the outputs of a single judge model. It requires no additional training, no extra annotations, and no ensemble of multiple judges, and it does not depend on any external tools.

**4. Clear structure of the paper (VUVV).**
Reviewers find that the overall structure of the paper is clear and logically organized, which makes the problem formulation, methodological components, and experimental conclusions easy to understand.

**The key additional experiments are as follows:**

**1. Additional experiments on Score–Comparison Inconsistency (VUVV, gSy2).**

* On the one hand, we compare three scoring methods across judge models: (a) a discrete 5-point baseline; (b) a continuous scoring obtained by applying softmax on the 5-point scale; (c) a continuous scoring obtained by applying softmax on the 100-point scale. The experiments show that simply applying softmax on top of the 5-point scale already substantially reduces Score–Comparison Inconsistency, indicating that continuous modeling of discrete scores is effective in itself. Extending the scale to 100 points further reduces inconsistency across different models and tolerance levels δ, showing that a finer-grained scale can yield additional improvements in consistency.

* On the other hand, to address reviewers’ concern that the single-scoring scale may depend on a particular tokenizer, we propose a scoring method based on the joint probability of the tens and ones digits, and conduct systematic evaluations on models such as gemma-2-27b-it and Qwen2.5-7B/14B. The results show that the scoring probabilities are highly concentrated over 0.9; the expected scores derived from the joint distribution align more closely with human annotations and achieve lower inconsistency rates in Score–Comparison Inconsistency than both discrete scoring and the G-Eval probabilistic baseline, indicating that this scheme is robust to tokenizer differences.

**2. Additional experiments on Pairwise Transitivity Inconsistency (nkL4, gSy2).**

* For a finer-grained analysis of non-transitivity types, we compute the proportions of inequality transitivity inconsistency and circular transitivity inconsistency across 11 judge models under three methods (Baseline, PPL-based, and Likelihood-aware Aggregation). The results show that TrustJudge reduces the proportions of both types of transitivity inconsistency.

* In terms of method comparison, we additionally reproduce PGED [1] and compare our single-judge TrustJudge setup against PGED's multi-judge configuration on the same tasks. The results indicate that TrustJudge outperforms PGED with 4 and 6 evaluators. We also discuss the fundamental difference in problem setting: our work focuses on improving the consistency of a single evaluator and does not rely on external models or tools. Moreover, in terms of computational complexity, TrustJudge requires only O(1) post-processing per sample, whereas PGED needs to construct and denoise a preference graph, whose complexity grows quadratically with the number of candidate responses.

We appreciate the committee's consideration and hope that these strengthened results are helpful for the decision-making process.

[1] Language Model Preference Evaluation with Multiple Weak Evaluators. arXiv:2410.12869.

---

### Meta-Review · Area_Chair_HTd1 · 2026-01-05

**Summary:**

This paper addresses two issues: (1) score-comparison inconsistency, where absolute ratings disagree with relative rankings; (2) pairwise intransitivity. The authors propose methods for resolving these disagreements.  They compare against G-Eval and a pairwise baseline that involves evaluating the pair in both directions.

Strengths:

- Interesting method for addressing inconsistencies in LLM-as-a-judge

- Formulation and evaluation of the inconsistency problem, unifying past work

- Detailed empirical analysis including ablations of the method.

Weaknesses:

1. Limited novelty over prior studies: As gSy2 points out, much of the ideas here have been explored previously. The system presented in this work is novel, but what is unclear is how much of that is due to fundamental theoretical contributions.

2. The main gains are on consistency metrics which the paper itself defines. While inconsistency is a broad problem for language models, the baselines and the proposed method are quite similar on win rate.

3. The method largely consists of heuristics that aren't backed by the theoretical justification. The theoretical justification establishes points about information loss which gSy2 argues are trivial (weakness 3). Major aspects like the PPL criterion and the 100-point scale come across as engineering decisions. Personally, I would advise the authors to tone down claims about how well-motivated these techniques are.

The paper explores a very interesting problem space, sets up a nice task with good motivation, and conducts worthy experiments, but doesn't feel like the best version of this paper that could be written in
terms of how these elements support a single, unified message.

**Reviewer Concerns:**

Reviewer gSy2 raises a number of relevant weaknesses.

Weakness 1: Novelty: the authors do not really rebut this point. They point to their existing evaluations but do not respond to the reviewer's specific assertions.  In particular, given what is known in the literature (e.g., intransitivity from Xu et al. "Investigating Non-Transitivity in LLM-as-a-Judge", and methods for using log-probability scoring, the difference from baselines seems like a minor implementation detail.

Regarding the sub-point about the comparison to Hu et al. "Language Model Preference Evaluation with Multiple Weak Evaluators", this method is only compared to in terms of NTR, not in terms of success (win rate).

More broadly, I think we should understand this paper as defining new metrics and then trying to improve those metrics over baselines, but only trying one fairly weak baseline method (the "pairwise" baseline). The baselines ultimately feel strawmannish: the authors' system wins not because we truly imagine it's the best approach for this, but because no other approach has been heavily optimized for this objective.

Weakness 2: Issues with the scale: I feel the paper does some sleight of hand here. There's a claim about the granularity of the scale being helpful. However, the authors aren't really showing that higher granularity is useful in general (which would require more experiments), just that 100-point is better than 5-point. This takes some theoretical analysis (which this reviewer isn't very convinced by) and then claims better empirical results, but I don't see this theory and this empiricism as very closely connected.

Weakness 4: "The core assumption that a sequence with lower PPL corresponds to a higher-quality response aligned with human preference is heuristic. It lacks a causal link and is not experimentally proven."
 I don't think this point is effectively rebutted.

**Reviewer Scores:**

In my judgment, I'm not gSy2 would be very convinced. In particular, points about novelty and the theoretical foundation aren't very well-addressed. However, the other reviewers are positive and would likely remain positive.

---

### Decision · Program_Chairs · 2026-01-26

Accept (Poster)